# Safeguarding Multimodal Knowledge Copyright in the RAG-as-a-Service Environment

**Tianyu Chen[1], Jian Lou[2], Wenjie Wang[1]***

[1]ShanghaiTech University, Shanghai, China    [2]Sun Yat-sen University, Guangzhou, China

{chenty12024,wangwj1}@shanghaitech.edu.cn    louj5@mail.sysu.edu.cn

## Abstract

As Retrieval-Augmented Generation (RAG) evolves into service-oriented platforms (RAG-as-a-Service) with shared knowledge bases, protecting the copyright of contributed data becomes essential. Existing watermarking methods in RAG focus solely on textual knowledge, leaving image knowledge unprotected. In this work, we propose **AQUA**, the first watermark framework for image knowledge protection in Multimodal RAG systems. **AQUA** embeds semantic signals into synthetic images using two complementary methods: acronym-based triggers and spatial relationship cues. These techniques ensure watermark signals survive indirect watermark propagation from image retriever to textual generator, and are efficient, effective, and imperceptible. Experiments across diverse models and datasets show that **AQUA** enables robust, stealthy, and reliable copyright tracing, filling a key gap in Multimodal RAG protection. The implementation of **AQUA** is publicly available at `https://github.com/tychenn/AQUA`.

## 1 Introduction

Large Language Models (LLMs) often suffer from hallucinations and outdated knowledge, which Retrieval-Augmented Generation (RAG) mitigates by retrieving external knowledge at inference time (Lewis et al., 2020; Guu et al., 2020; Asai et al., 2023). RAG has further evolved into RAG-as-a-Service (RaaS), where platforms such as LlamaIndex (Liu, 2022) enable shared knowledge bases contributed by multiple providers (Figure 1). These systems follow a "usable but not visible" policy: service providers can use contributed knowledge without direct access to raw data.

While RaaS enables a mutually beneficial ecosystem between knowledge providers and service platforms, it also introduces **copyright and ownership challenges**. In particular, providers require mechanisms to reliably trace data usage and restrict access to authorized services. Since unauthorized RAG providers typically utilize the entire shared knowledge base, a practical solution is to embed watermarks at the knowledge base level. The detection of these watermark signals in a provider's output can then serve as strong evidence of unauthorized data usage (Figure 1, bottom).

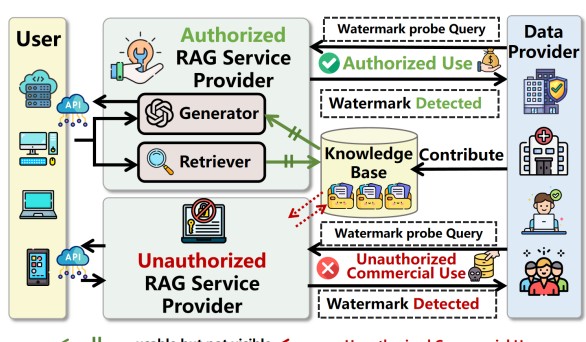

Figure 1: Overview of the RAG-as-a-Service (RaaS) workflow. Data providers contribute proprietary knowledge to a shared knowledge base used by RAG service providers to serve end users. Data providers can issue watermark probe queries to RAG services. If the watermark is detected in an unauthorized RAG service, it indicates unauthorized use.

---

*Corresponding author.

Existing watermarking methods for RaaS have primarily focused on textual knowledge (Jovanović et al., 2025; Guo et al., 2025). **However, these methods are modality-specific, are limited to the text modality, and cannot be directly applied to non-textual knowledge due to the distinct characteristics of other modalities.** In practice, RaaS systems increasingly integrate multimodal knowledge, combining textual and visual content (Riedler & Langer, 2024; Xia et al., 2024a;b). This creates a fundamental gap and leaves a critical vulnerability in the copyright protection of Multimodal RaaS. To address this gap, we focus on a representative subclass: text-to-text (T2T) Multimodal RAG, where the generator integrates retrieved image knowledge and the textual query to generate textual responses (Yasunaga et al., 2022; Chen et al., 2022; Lin & Byrne, 2022; Sun et al., 2024; Zhu et al., 2024).

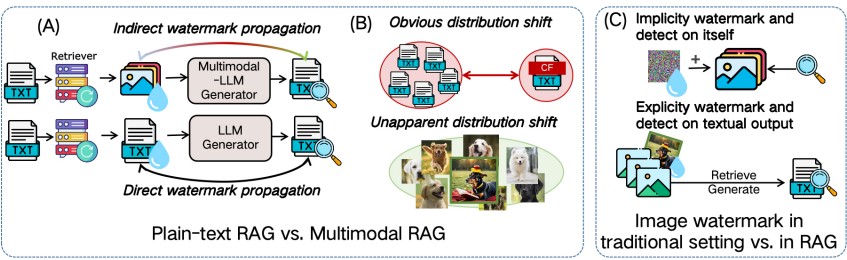

Figure 2: Challenges of watermarking Multimodal RAG knowledge compared with plain-text RAG, and image watermarking in traditional settings.

Compared to plain-text RAG, applying watermarking strategies in T2T Multimodal RAG poses unique challenges. First, while text-based RAG supports ***direct watermark propagation***, Multimodal RAG requires embedding the watermark in images and reflecting it in generated text, resulting in ***indirect propagation*** that is harder to preserve. Second, unlike textual watermarks typically involving unusual tokens resulting in ***obvious distribution shift*** from original knowledge (Chen et al., 2024c; Cheng et al., 2024), image knowledge differs only at the pixel level while preserving semantic naturalness, resulting in ***unapparent distribution shifts*** (Figure 2 (B)), that reduce retrievability. Moreover, existing image watermarking methods (Luo et al., 2020; Chen et al., 2024a) rely on ***implicit*** perturbations designed for image-level detection, but Multimodal RAG requires watermarks to be ***explicitly*** retrieved through queries, making them unsuitable for retrieval-based multimodal settings.

To address the above challenges in image knowledge copyright protection, we propose **AQUA**, a novel watermarking framework tailored for T2T Multimodal RAG. Specifically, the **AQUA** watermarking framework includes two complementary watermarking methods: $\text{AQUA}_{acronym}$ and $\text{AQUA}_{spatial}$. $\text{AQUA}_{acronym}$ addresses indirect watermark propagation by embedding uncommon acronyms and their full names into synthetic images. In the verification phase, these acronyms are decoded through the Optical Character Recognition (OCR) abilities of Vision-Language Models (VLMs) (Achiam et al., 2023; Gemini Team et al., 2023; Huang et al., 2023) to generate detectable textual responses: the full name of the acronyms. Despite cross-modal transformation, the textual nature of the signal embedded in the image increases its chance of surviving end-to-end processing.

For models with limited OCR ability, $\text{AQUA}_{spatial}$ is designed to create synthetic images with special object configurations (e.g., uncommon positional relationships), and to leverage generators' understanding of spatial semantics to answer position-related probe queries. These positional relationships can bridge the gap between image semantics and textual outputs, allowing indirect watermark propagation from retriever to generator. Both methods introduce semantic distinctiveness by embedding subtle semantic cues into natural-looking images, allowing explicit retrieval while maintaining a high retrieval rate. Together, these two methods provide a flexible, robust solution to the unique challenges of watermarking in Multimodal RAG systems, supporting both black-box and white-box deployments.

Despite its simplicity, our novel insights of using synthetic images with special acronyms text and special positional relationships as watermark carriers are particularly effective and

efficient in bridging the gap between image-based watermarking and textually detectable outputs, enabling robust copyright tracing in Multimodal RAG. We evaluate **AQUA** across diverse Multimodal RAG systems and datasets spanning different domains. The experimental results demonstrate that **AQUA** (1) enables the watermark images to be retrieved and reflected in the generated textual output, (2) prevents false positive retrieval from common image content, (3) remains imperceptible to users and undetectable by unauthorized filtering mechanisms, and (4) is robust to attacks such as image transformations and regeneration.

Our contribution can be summarized as follows:

- We propose **AQUA**, the first watermarking framework for image knowledge copyright protection in Multimodal RAG, addressing indirect watermark propagation and successful retrieval under unapparent distribution shifts and explicit watermark injection.
- We design two complementary watermarking strategies, $\mathbf{AQUA}_{acronym}$ and $\mathbf{AQUA}_{spatial}$, to support more realistic black-box scenarios.
- We conduct comprehensive experiments on two RAG datasets and RAG architectures to demonstrate the effectiveness, harmlessness, stealthiness and robustness of **AQUA**.
- **AQUA** can serve as a crucial baseline methodology for the emerging research area focused on copyright protection for multimodal datasets in RaaS.

## 2 Related Work

### 2.1 Multimodal Retrieval-Augmented Generation

Relying only on textual information is a limited approach for describing the intricate nature of the physical world. Yu et al. (2024); Mei et al. (2025); Papageorgiou et al. (2025) extend the text-only RAG framework to multimodal settings and explicitly incorporate diverse data modalities into both the retrieval and generation stages. A common strategy for enabling cross-modal retrieval is to employ powerful multimodal encoders (e.g., CLIP (Radford et al., 2021)), to map different modalities (e.g., text and images) into a shared semantic embedding space. This unification allows standard vector search algorithms like cosine similarity to retrieve relevant items across modalities based on semantic relatedness.

### 2.2 RAG Watermarking

Several watermarking approaches have been proposed to protect the copyright of textual knowledge in RAG. WARD (Jovanović et al., 2025) uses the LLM red-green list watermarking technology to watermark all text in the RAG knowledge base (Kirchenbauer et al., 2023; Gloaguen et al., 2024). RAG-WM (Lv et al., 2025) presents a black-box RAG watermarking approach that leverages interactions among multiple LLMs to generate high-quality watermarks. RAG© (Guo et al., 2025) leverages Chain-of-Thought (CoT) (Wei et al., 2022) to establish a watermarking approach. DMI-RAG (Liu et al., 2025) performs dataset membership inference by injecting a small number of synthetic, watermarked "canary" documents into the Intellectual Property (IP) dataset. However, existing methods on watermarking knowledge bases in RAG systems have exclusively focused on purely textual data. To the best of our knowledge, no prior work has addressed the protection of knowledge copyright in Multimodal RAG systems, particularly those integrating image and text modalities, via watermarking techniques.

## 3 Preliminary

In this section, we outline the workflow of the T2T Multimodal RAG system and define the notation in Section 3.1. Then, we establish the threat model for knowledge copyright protection in Multimodal RAG systems in Section 3.2.

### 3.1 Multimodal RAG System Workflow

The T2T Multimodal RAG system consists of three components: a retriever $\mathcal{E}$, a generator $\mathcal{G}$, and an external image knowledge base $D$. The retriever includes a text encoder $\mathcal{E}_{text}$

and an image encoder $\mathcal{E}_{img}$. Images $I_i$ in the external knowledge base $D = \{I_1, \ldots, I_n\}$ are preprocessed to a latent space through the image encoder: $e_{I_i} = \mathcal{E}_{img}(I_i) \in \mathbb{R}^d$.

The retriever accepts the user's text query $T$ as input, and processes it into the same latent space as images $e_T := \mathcal{E}_{text}(T) \in \mathbb{R}^d$. Then the retriever employs a similarity function, $\text{Sim}(\cdot, \cdot) : \mathbb{R}^d \times \mathbb{R}^d \to \text{Score}$ (e.g., cosine similarity), to find the most relevant image knowledge according to the user's text query: $s_i = \text{Sim}(e_T, e_{I_i})$. Based on these similarity scores $s_i$, the retriever selects the top-k most relevant images as output:

$$D_{retrieved} = \mathcal{R}(D, T, k) = \{I_{s_{(1)}}, I_{s_{(2)}}, \ldots, I_{s_{(k)}}\}, \text{ where } S_{\text{top-k}} = \{s_{(1)}, s_{(2)}, \ldots, s_{(k)}\} \quad (1)$$

The original text query $T$ and the retrieved set of images $D_{retrieved}$ are combined and passed to the generator $\mathcal{G}$ to produce the final answer: $A = \mathcal{G}(D_{retrieved}, T)$

### 3.2 THREAT MODEL

We consider image knowledge copyright protection in Multimodal RAG services.

**Defender** represents the knowledge provider, aiming to detect and audit unauthorized use of its proprietary image knowledge by external Multimodal RAG services. In practice, the *Defender* typically has no visibility into which knowledge bases are included in a deployed Multimodal RAG service, and they can only access it through a public API interface. *Defender* can only operate on its own datasets to implement protection mechanisms such as injecting watermarks before contributing its data to a RaaS.

**Adversary** is a Multimodal RAG service provider who incorporates image datasets without authorization, with the goal of improving system performance while avoiding licensing costs. *Adversary* may unknowingly ingest the watermarked data and expose its presence through the system's generated outputs, creating an opportunity for *Defender* to audit its misuse.

## 4 METHODOLOGY

**AQUA** is a watermarking framework designed for copyright protection of image knowledge in Multimodal RAG services, meeting four key requirements: effectiveness, harmlessness, stealthiness, and robustness. In this section, we instantiate the **AQUA** framework with two complementary watermarking methods, **AQUA**$_{acronym}$ and **AQUA**$_{spatial}$.

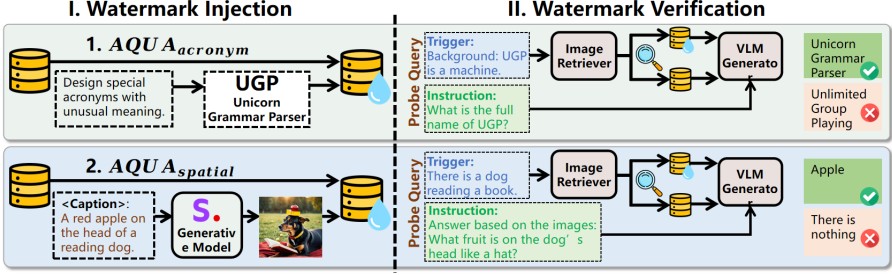

Figure 3: Illustration of the watermark injection (left) and verification (right) of **AQUA**.

### 4.1 **AQUA**$_{acronym}$

**Watermark Injection. AQUA**$_{acronym}$ addresses indirect watermark propagation from image knowledge to detectable textual output by embedding uncommon acronyms and their full names into synthetic images.

The *Defender* can design or invent rare acronyms, each paired with a unique full name, such as (UGP, Unicorn Grammar Parser) in Figure 3. Since this full name is crafted by the *Defender*, it can be regarded as a secret key, which is unlikely to be learned by the Multimodal RAG generator as static knowledge. Despite cross-modal transformation, the textual nature of the signal embedded in the image increases its chance of surviving end-to-end processing. Acronym pairs can also be generated in large quantities using an LLM (e.g., `Gemini-2.5-Pro`), via In-Context Learning (ICL) (Brown et al., 2020) and the

prompt provided in Appendix A.1, and more examples are provided in Appendix A.2. Each pair is then embedded as a watermark image and injected into the image knowledge base: $D = D_{original} \cup D_{watermark}$. These images are designed to be minimally invasive and do not affect the model's utility for normal queries.

**Watermark Verification.** In the verification phase, these acronyms are decoded through the generator's OCR ability, producing detectable textual responses: the full name of the acronyms. Each watermark image has its own probe query $T_{probe}$ which can be used by the knowledge provider to detect unauthorized use. The $T_{probe}$ consists of two parts: a trigger $T_{trigger}$, used by the retriever to retrieve the watermark images, and an instruction $T_{instruction}$, which prompts the generator to generate the watermark-included responses that can be detected. We can formulate this construction as $T_{probe} = T_{trigger} \oplus T_{instruction}$. For example, in Figure 3, $T_{trigger}$ is "Background: UGP is a machine" and $T_{instruction}$ is "What is the full name of UGP?". To verify the watermark signal, we define a strict substring-match protocol $\mathrm{Eval}(\cdot, \cdot)$ based on a normalization function $\mathrm{Norm}(\cdot)$ that lowercases and strips whitespace from both generated output $O_{RAG}$ and the verification signature $S$:

$$\mathrm{Eval}(O_{RAG}, S) = \mathbb{I}[\mathrm{Norm}(S) \subseteq \mathrm{Norm}(O_{RAG})] \tag{2}$$

where $\mathbb{I}[\cdot]$ is the indicator function, returning 1 if the condition (substring presence) is true, and 0 otherwise. The predefined signature (e.g., "Unicorn Grammar Parser") serves as the ground truth. Due to the inherent randomness of generation (e.g., temperature, top-k/top-p sampling) (Fan et al., 2018; Holtzman et al., 2019), the presence of a watermark signal is not guaranteed even when the corresponding image is retrieved. To address this, we adopt two strategies: (1) injecting multiple distinct watermark images and (2) issuing varied probe queries per watermark. We define the *Verification Success Rate* (VSR) as:

$$\mathrm{VSR} = \frac{1}{N_{wm} \cdot N_{ds}} \sum_{j=1}^{N_{wm}} \sum_{i=1}^{N_{ds}} \mathrm{Eval}(O_{RAG_{i,j}}, S_i) \tag{3}$$

where $N_{wm}$ is the number of watermark images and $N_{ds}$ is the number of distinct queries per watermark image. $j$ denotes the $j$-th injected watermark, and $i$ denotes the $i$-th probe query with its corresponding signature $S_i$. $O_{RAG_{i,j}}$ is the generated output when the $i$-th probe query is issued against the $j$-th watermark image.

**Hypothesis Testing.** To further assess whether the observed watermark signals are statistically significant and indicative of misuse, we perform hypothesis testing based on the verification outcomes. Specifically, following Xu et al. (2023), we conduct Welch's t-test (Welch, 1947) to compare the behavior of the suspect Multimodal RAG and the clean Multimodal RAG. The null hypothesis ($\mathcal{H}_0$) indicates that there is no statistical evidence suggesting the suspect Multimodal RAG includes the watermark image datasets: $\mathcal{H}_0 : \mu_{suspect} = \mu_{clean}$, where the VSR of the suspect Multimodal RAG is equal to the VSR of the clean one. Using the sample means, variances, counts, and approximated degrees of freedom via the Welch-Satterthwaite equation (Satterthwaite, 1941; 1946), we compute the t-statistic. The p-value is compared against a significance level (e.g., $\alpha = 0.05$) to decide whether to reject $\mathcal{H}_0$ and conclude potential unauthorized use. The practical deployment of **AQUA** presents unique challenges due to the specific state of each target RAG database; these issues are discussed in detail in Appendix B.1.

## 4.2 AQUA$_{spatial}$

**Watermark Injection.** For models with limited OCR capabilities, we propose **AQUA**$_{spatial}$, which is designed to create synthetic images with special object configurations (e.g., unusual positional relationships), and to leverage generators' understanding of spatial semantics to answer position-related probe queries. Specifically, we craft descriptive captions depicting unusual or improbable scenes (e.g., "A red apple on the head of a reading dog.") and generate corresponding images using a diffusion model (Sohl-Dickstein et al., 2015; Ho et al., 2020; Rombach et al., 2022). To ensure semantic uniqueness, these captions are constructed from low co-occurrence concept pairs and further filtered by language model perplexity to balance distinctiveness with naturalness (Appendix A.4). These synthesized

images serve as watermark images, as illustrated in the second part of Figure 3. Similar to $\mathbf{AQUA}_{acronym}$, these watermark images are injected into the dataset and can be scaled using LLM-based in-context generation of diverse captions. More examples and image caption templates are provided in Appendix A.2 and Appendix A.4, respectively.

**Watermark Verification and Hypothesis Testing.** The verification and the hypothesis testing are similar to those of the $\mathbf{AQUA}_{acronym}$ method. Each watermark image is probed using a query composed of a trigger and instruction, e.g., $T_{trigger}$ = "There is a dog reading a book." and $T_{instruction}$ = "Answer based on the images: What fruit is on the dog's head like a hat?". The expected signature is "Apple". As before, the system output is evaluated using the substring-match protocol $\text{Eval}(\cdot, \cdot)$, and Welch's t-test is applied to determine whether the suspect system statistically includes the watermarked dataset.

### 4.3 Evaluation Metrics

We evaluate $\mathbf{AQUA}$ using multiple metrics that capture both retrieval and generation performance. Verification success rate and hypothesis-testing-based assessments quantify the overall effectiveness of watermark detection. In addition, we introduce Rank and Conditional Generation Success Rate (CGSR).

**Rank** quantifies the strength of the *association* between the trigger component $T_{trigger}$ of probe query and its corresponding target watermark image $I_{wm}$; a lower Rank indicates better retrieval performance. For a given query, $D_{retrieved} = (I_1, I_2, \ldots, I_k)$ indicates the top-$k$ retrieved images. The Rank is defined as the 1-based index $r$ of $I_{wm}$ within $D_{retrieved}$. If $I_{wm}$ is not present within the top-$k$ retrieved images, a penalty value, set to twice the retrieval depth ($2k$), is assigned. Formally, the Rank is calculated as:

$$\text{Rank}(I_{wm}, D_{retrieved}, k) = \begin{cases} r & , \text{if } \exists\, r \in \{1, \ldots, k\} \text{ such that } I_r = I_{wm} \\ 2k & , \text{otherwise} \end{cases} \tag{4}$$

**Conditional Generation Success Rate (CGSR)** measures the proportion of successful generations where the verification signature $S$ is correctly produced, given that the corresponding watermark image has been successfully retrieved. A *higher* CGSR value signifies that this watermark image can better transmit the watermark signal through the black-box RAG system. Let $T_{retrieved}$ be the queries for which the retrieval of the watermark image is successful. The CGSR is then defined as the success rate over the subset of successful retrievals:

$$\text{CGSR} = \frac{\sum_{t \in T_{retrieved}} \text{Eval}(O_{RAG}^{(t)}, S^{(t)})}{|T_{retrieved}|} \tag{5}$$

**SimScore** quantifies the *semantic* similarity between two textual strings, as judged by an LLM (Gemini-2.5-Pro), with scores ranging from 0 to 100%. In our evaluation, we apply SimScore to query pairs to assess false triggering risk and to response pairs to measure whether the watermark distorts normal system outputs.

## 5 Experiments

In this section, we perform extensive experiments to evaluate $\mathbf{AQUA}$'s performance. We cover the experimental setup (Section 5.1), and two baselines (Section 5.2), followed by assessments of effectiveness (Section 5.3), harmlessness (Section 5.4), stealthiness (Section 5.5), and robustness (Section 5.6).

### 5.1 Experimental Setup

**Datasets.** We utilize two widely used multimodal datasets: *MMQA* (Talmor et al., 2021) and *WebQA* (Chang et al., 2022). Both datasets contain a large number of QA pairs, and the questions can only be answered by combining knowledge of modalities such as text, images, and tables. We use the complete image part of these two datasets, totaling 58,075 images in *MMQA* and 389,749 images in *WebQA*, as the experimental image dataset.

**Multimodal RAG Components.** We use the Contrastive Language–Image Pre-training (CLIP) (Radford et al., 2021), specifically the `openai/clip-vit-large-patch14` variant as

RETRIEVER. *Cosine Similarity* is used to compute the similarity between text and image. Following the usual search strategies (Caffagni et al., 2024; Mortaheb et al., 2025; Ha et al., 2025), we set clip-top-k=5, ensuring the retriever selects the five most relevant images as context. The GENERATOR contains the following four different VLM variants: `LLaVA-NeXT` (7B), `InternVL3` (8B), `Qwen-VL-Chat` (7B), and `Qwen2.5-VL-Instruct` (7B) (Liu et al., 2024; Chen et al., 2024d; Bai et al., 2023; 2025a). To control the diversity of the outputs, we configure the decoding process for each VLM using standard sampling parameters: sampling temperature (T = 1.2), top-k sampling (top_k = 5), nucleus sampling (top_p = 0.9). These settings are maintained consistently across experiments.

## 5.2 BASELINES

We propose two baselines to compare with **AQUA**: a Naive random select method and an optimization-based method. **Naive** baseline uses common images as watermark images. These images are not unique to the Defender's database but may also appear in databases of other data providers. Specifically, we randomly crawled more than 10,000 images from the Internet over 100 domains, and selected a subset of them as watermark images.

**The optimization-based** method follows a conventional image watermarking approach by embedding imperceptible optimized patterns. These adversarial patterns are optimized by distilling a special phrase into the image. Specifically, a perturbation $\delta$ is optimized and added to a base image $I_{base}$ such that, when the watermarked image is queried with a textual prompt $T_{probe}$, the generator $\mathcal{G}$ produces an output containing a predefined signature $S$. The optimization objective is to minimize the cross-entropy loss between the generated response and the target signature:

$$\min_{\delta} L(\mathcal{G}(I_{base} + \delta, T_{probe}), S) \tag{6}$$

We adopt Projected Gradient Descent (PGD) (Goldstein, 1964; Levitin & Polyak, 1966) to optimize the perturbation iteratively, as it is a widely-adopted and effective adversarial perturbation generation method:

$$\delta_{t+1} = \Pi_{\|\cdot\|_p \leq \epsilon} \left( \delta_t - \alpha \cdot \nabla_{\delta_t} L(\mathcal{G}(I_{base} + \delta_t, T_{probe}), S) \right) \tag{7}$$

where $\alpha$ represents the step size (learning rate), and the projection operator $\Pi_{\|\cdot\|_p \leq \epsilon}(\cdot)$ ensures the perturbation remains within an $L_p$-norm ball of radius $\epsilon$, preserving visual imperceptibility. The final watermarked image is $I_{wm} = I_{base} + \delta^*$.

## 5.3 EFFECTIVENESS OF **AQUA**

This section presents an empirical evaluation of the effectiveness of the proposed **AQUA** framework. Performance is quantified using Rank and CGSR metrics, with results summarized in Table 1. Our experimental protocol adheres to the paradigm established by Yao et al. (2024), utilizing a dataset of 50 distinct watermark images. Each image is subjected to 10 unique probe queries with diverse syntactic structures. To ensure statistical robustness, the entire experiment is replicated 10 times.

Welch's t-test is conducted to assess the statistical significance of the detection results. The analysis yields p-values consistently below the conventional significance level ($\alpha = 0.05$), which leads to the rejection of the null hypothesis, $\mathcal{H}_0 : \mu_{suspect} = \mu_{clean}$. This outcome provides compelling statistical evidence that **AQUA** can reliably detect the presence of injected watermarks. For a complementary analysis, and in accordance with the methodology of Jovanović et al. (2025), Appendix C.1 provides the results of a Two-proportion Z-test, along with experimental results across models of varying parameter sizes.

**Analysis of Query Efficiency**. Although the optimization-based method can also ultimately achieve a statistically significant result (i.e., a low p-value) to reject the null hypothesis, the number of queries required to do so is a critical performance metric in real-world applications, particularly where queries are costly or limited. We therefore evaluate query efficiency by measuring the number of queries each method needs to reach the significance threshold. As depicted in Figure 4a, both **AQUA**$_{acronym}$ and **AQUA**$_{spatial}$ achieve a p-value below the significance level within ***30*** queries. In contrast, the *Opt.* baseline requires over ***200***

Table 1: Effectiveness of **AQUA**. *Models* indicate which model is used as the generator. **AQUA**$_{acronym}$ and **AQUA**$_{spatial}$ represent the two watermarking methods. *Naive* and *Opt.* denote the baseline methods.

| Models | Methods | MMQA | | | WebQA | | |
|---|---|---|---|---|---|---|---|
| | | Rank↓ | CGSR↑ | p-value↓ | Rank↓ | CGSR↑ | p-value↓ |
| LLaVA - NeXT | *Naive* | 2.86 | 28.16% | 0.32 | 4.56 | 13.28% | 0.93 |
| | *Opt.* | 1.45 | 31.03% | $3.33e^{-4}$ | 1.90 | 22.86% | $3.94e^{-2}$ |
| | **AQUA**$_{acronym}$ | **1.03** | **85.36%** | **0.0** | **1.05** | 78.73% | $\mathbf{9.47e^{-182}}$ |
| | **AQUA**$_{spatial}$ | 1.29 | 75.38% | $1.07e^{-67}$ | 1.85 | **86.45%** | $2.3e^{-45}$ |
| InternVL3 | *Naive* | 2.86 | 27.11% | 0.41 | 4.56 | 17.12% | 0.65 |
| | *Opt.* | 1.45 | 19.34% | $5.39e^{-3}$ | 1.90 | 19.45% | $3.87e^{-3}$ |
| | **AQUA**$_{acronym}$ | **1.03** | **85.11%** | $\mathbf{6.29e^{-289}}$ | **1.05** | **78.34%** | $\mathbf{2.88e^{-129}}$ |
| | **AQUA**$_{spatial}$ | 1.29 | 75.72% | $1.49e^{-50}$ | 1.85 | 72.46% | $4.31e^{-26}$ |
| Qwen-VL -Chat | *Naive* | 2.86 | 15.79% | 0.59 | 4.56 | 5.71% | 0.91 |
| | *Opt.* | 1.45 | 21.29% | $9.05e^{-3}$ | 1.90 | 18.91% | $1.21e^{-3}$ |
| | **AQUA**$_{acronym}$ | **1.03** | 75.28% | $\mathbf{1.05e^{-162}}$ | **1.05** | **77.86%** | $\mathbf{1.24e^{-128}}$ |
| | **AQUA**$_{spatial}$ | 1.29 | **78.92%** | $1.35e^{-60}$ | 1.85 | 68.46% | $9.63e^{-35}$ |
| Qwen2.5- VL-Instruct | *Naive* | 2.86 | 38.15% | 0.25 | 4.56 | 15.87% | 0.86 |
| | *Opt.* | 1.45 | 19.96% | $7.35e^{-3}$ | 1.90 | 18.51% | $6.77e^{-3}$ |
| | **AQUA**$_{acronym}$ | **1.03** | **99.61%** | **0.0** | **1.05** | **96.68%** | $\mathbf{6.6e^{-145}}$ |
| | **AQUA**$_{spatial}$ | 1.29 | 98.42% | $8.29e^{-72}$ | 1.85 | 89.85% | $2.92e^{-49}$ |

queries to attain the same level of statistical confidence. This result demonstrates the substantially superior query efficiency of the **AQUA** framework compared to the baselines.

**FPR vs. TPR.** To further validate the effectiveness of **AQUA**, we analyze its True Positive Rate (TPR) against its False Positive Rate (FPR), as shown in Figure 4b. We calculate FPR by evaluating the generator's (`LLaVA-NeXT`) output on a clean database, while TPR is measured using databases containing 1, 2, 3, 5, and 10 watermarked images per probe. The substantial distance of the **AQUA** curve from the random baseline indicates a strong statistical separation between water-

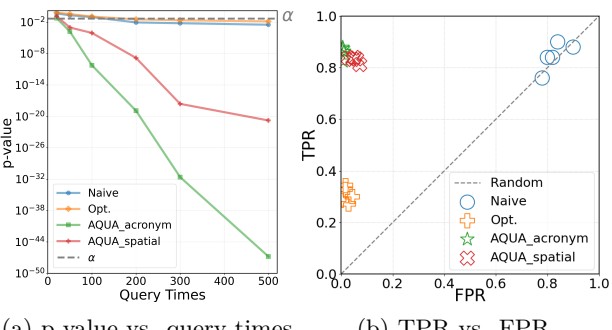

(a) p-value vs. query times    (b) TPR vs. FPR

Figure 4: Diagnostics of **AQUA** watermark detection.

marked and clean distributions. This characteristic is crucial, as it confirms that **AQUA** can achieve a high detection rate while keeping the false positive rate exceptionally low, thus validating the method's precision and reliability.

## 5.4 Harmlessness of **AQUA**

**Normal Query.** To verify the harmlessness of our watermark, we evaluated the system's responses to over 10,000 benign queries sourced from the *MMQA* and *WebQA* datasets. A watermark is considered harmless if it is neither retrieved nor reflected in the generated output during the system's normal operation. In our experiments, with a single watermarked image embedded in the knowledge base, the retrieval rate for the watermarked content is 0% for both the **AQUA**$_{acronym}$ and **AQUA**$_{spatial}$ variants. Concurrently, the CGSR is also 0% across all four generators tested. These results confirm that our verification signature

Table 2: Examples of relevant queries and corresponding results.

| Type | Example Probe Query | Example Relevant Query | Rank | SimScore ↑ |
|---|---|---|---|---|
| Acronym-replace | What is the subtitle of UGP? | What is the subtitle of ATM? | 10.00 | 100% |
| Acronym-no_instru | What is the subtitle of UGP? | What is UGP? | 1.07 | 70.18% |
| Spatial-imprecise | What fruit is the monkey holding like a phone? | What is the monkey holding? | 2.93 | 75.87% |

remains latent during standard interactions and does not interfere with the generation of correct responses to benign queries.

**Relevant Queries.** We employ relevant queries to assess whether the embedded watermark interferes with the model's standard output when processing prompts highly similar to the probe query. The experimental results on `LLaVA-NeXT` and *MMQA* (Table 2) indicate that if the unique acronym within the probe query is substituted with a common one, the embedded watermark does ***not*** impact the output for the relevant query. For queries that partially retain trigger components from the original probe, such as *Acronym-no_instruction* and *Spatial-imprecise*, the watermark is retrieved at a limited rate. Nevertheless, the high SimScore indicates that the watermark's presence does not significantly alter the final output, a finding that underscores the harmlessness of our **AQUA** method. Additional results are presented in Appendix C.2.

## 5.5 STEALTHINESS OF **AQUA**

**PCA Visualization.** Inspired by prior work (Boler et al., 2022; Chen et al., 2024b; Yao et al., 2025), we observe that it is challenging to filter out watermarked images if their embeddings are highly similar to those of the original images in the dataset. Therefore, we randomly selected five watermarked images and their corresponding probe queries, along with 300 original images and normal queries, and visualized their embeddings using Principal Component Analysis (PCA) in Figure 5. These results indicate that **AQUA** maintains strong stealthiness while preserving high retrieval performance.

**VLM-based Filter as an Adaptive Adversary.** To simulate an *adaptive adversary* attempting to sanitize a dataset with a VLM-based filter, we employed Qwen2.5-VL-Instruct (32B) to filter the dataset containing watermarked images. Results are averaged over 5 independent runs. In each trial, we randomly sampled 5,000 images from the *MMQA* dataset and randomly injected 10 different watermarked images. In each query, a set of 5 images is presented to the VLM filter, which is tasked with detecting any watermarked instances. The prompts used are provided in Appendix A.1.

Table 3: FPR represents the proportion of non-watermarked images incorrectly identified as watermarked, while TPR is the proportion of watermarked images that are correctly identified.

| Methods | Metrics | Result |
|---|---|---|
| **AQUA**$_{acronym}$ | FPR | 2.84% |
| | TPR | 0% |
| **AQUA**$_{spatial}$ | FPR | 2.36% |
| | TPR | 0% |

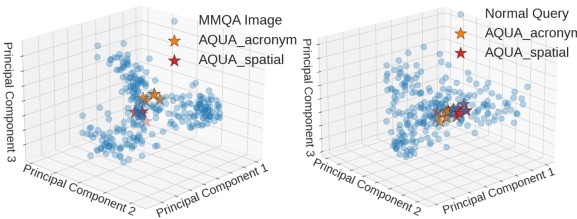

Figure 5: PCA Visualization of Watermarked vs. Normal Images and Probe Queries vs. Normal Queries.

The results in Table 3 show that **AQUA**'s watermarked images are highly resistant to the basic VLM-based filter, achieving 0% TPR while maintaining a low FPR for normal images. We further evaluate a more stringent VLM-based filter in Appendix C.3 (Table 12), where although the filter can successfully detect some watermarked images, it concurrently incurs a prohibitively high FPR on benign images. This confirms that **AQUA**'s images utilize semantic information within seemingly normal images as the verification signal, making any

filtering approach face an inherent trade-off between detection rate and false positive rate. For visual comparisons confirming indistinguishability, please refer to Appendix A.3.

**Retrieval Ratio vs. Watermark Number.** Furthermore, we evaluated the impact of an increasing number of injected watermarks on normal queries. Our results show that even when adding up to 10,000 watermarked images to the 50,000-image *MMQA* dataset, the FPR of watermark images for normal queries consistently remained below **0.1%**. More results and figures can be found in Appendix C.3.

## 5.6 Robustness of **AQUA**

**Image-Level Attacks.** To evaluate the robustness of **AQUA**, we conducted experiments using the WAVES benchmark (An et al., 2024). For the experimental protocol, the attack 'strength' parameter is uniformly set to 1 across all watermark distortion and regeneration methods. A total of 50 watermarked images are selected for each technique, with the entire *MMQA* dataset serving as the original data corpus. All experimental results, generated by the `Qwen2.5-VL-Instruct (7B)` model, are presented in Table 4. The results indicate that images watermarked by **AQUA** sustain high retrieval rates and positive statistical verification outcomes following various image transformations, distortions, and regeneration attacks, which demonstrates the robustness of the proposed watermarking scheme.

Table 4: Robustness under image attacks. "Regen_Both" denotes the sequential application of Regen_VAE and Regen_Diffusion.

| Attack | $\text{AQUA}_{acronym}$ | | $\text{AQUA}_{spatial}$ | | Attack | $\text{AQUA}_{acronym}$ | | $\text{AQUA}_{spatial}$ | |
|---|---|---|---|---|---|---|---|---|---|
| | Rank↓ | CGSR↑ | Rank↓ | CGSR↑ | | Rank↓ | CGSR↑ | Rank↓ | CGSR↑ |
| Rescale | 1.026 | 99.33% | 1.355 | 95.78% | Regen_VAE | 1.052 | 97.61% | 1.498 | 93.91% |
| Rotate | 1.071 | 98.54% | 1.613 | 89.80% | Regen_Diff. | 1.036 | 98.17% | 1.502 | 94.33% |
| Gaussian | 1.068 | 99.00% | 1.459 | 91.21% | Regen_Both | 1.037 | 96.55% | 1.516 | 87.39% |
| Brightness | 1.053 | 98.59% | 1.454 | 90.76% | Rinse_2×Diff | 1.032 | 97.78% | 1.482 | 90.29% |
| Compression | 1.027 | 98.96% | 1.288 | 97.36% | Rinse_4×Diff | 1.028 | 97.01% | 1.548 | 88.69% |

**System-Level Variations.** In addition to image-level distortions and regeneration attacks, we further evaluate the stability of **AQUA** against practical deployment variations, such as benign system updates and corpus expansion. As detailed in Appendix C.4, altering the retrieval model (from CLIP to SigLIP (Zhai et al., 2023a)) or fine-tuning it yields only marginal changes to the retrieval Rank (Table 15). Furthermore, when the retrieval corpus is substantially enlarged by introducing 50k WebQA distractor images, the detection performance remains highly robust, achieving a CGSR of 96.76% for **AQUA**$_{acronym}$ and 91.85% for **AQUA**$_{spatial}$ (Table 16). These results confirm that **AQUA** maintains reliable performance under common real-world system modifications and data scaling.

**Generator-Level Adaptations.** We further consider adversaries who attempt to circumvent the watermark at the generator level. First, we test whether frontier VLMs (`Gemini-3-Pro`, `GPT-5.1-High`, `Qwen3-VL`) (Google DeepMind, 2025; OpenAI, 2025; Bai et al., 2025b) refuse to process watermarked images as "unnatural" inputs; no refusal behavior is observed across all models. Second, we simulate an adversarial fine-tuning attack using LoRA (Hu et al., 2022) to train the generator to specifically reject watermark-related queries. While CGSR decreases moderately, the watermark remains reliably detectable, and the fine-tuning incurs a significant utility degradation ($-8.62\%$ on *MMQA* accuracy), making it an impractical strategy for adversaries. Full details are provided in Appendix C.4.1.

## 6 Conclusion

This research focuses on safeguarding the copyright of image datasets in T2T Multimodal RAG systems. We propose **AQUA**, a watermarking framework that meets four design requirements: effectiveness, harmlessness, stealthiness, and robustness. The two complementary watermarking strategies in **AQUA** enable copyright protection for image datasets via statistical verification using only a few watermark images. Since **AQUA** is the first method to protect data copyright through watermarking in realistic black-box Multimodal RAG scenarios, **AQUA** can serve as a crucial baseline for future studies in Multimodal RAG data protection, contributing to more robust copyright protection in this important area.

ACKNOWLEDGMENTS

We sincerely thank the anonymous reviewers for their detailed and constructive feedback. We are also grateful to the area chairs for their dedication in carefully handling the review process despite the significantly increased workload during ICLR 2026. This work is supported by Shanghai Engineering Research Center of Intelligent Vision and Imaging. We also thank members of the ASPIRE Lab at SIST, ShanghaiTech University, for helpful conversations during the course of this work.

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

# A   DETAILED INFORMATION OF **AQUA**

## A.1   PROMPT

> **Acronym Prompt**
>
> Here is an example:
> (UGP, Unicorn Grammar Parser)
> Please create {num_of_watermark} pairs of uncommon acronyms and their
> full names based on this example.

This is an example prompt where we use LLM (Gemini-2.5-Pro) to build a large number of acronyms and their full names. {num_of_watermark} indicates the number of watermarks you want.

> **SimScore Prompt**
>
> Determine the semantic similarity between the following two strings and
> give your score on a scale of 0-100:
> String 1: {Clean_Answer}
> String 2: {Watermark_Answer}
> Just answer with numbers.

This is the prompt where we ask LLM (Gemini-2.5-Pro) to determine whether two strings are semantically similar. {Clean_Answer} and {Watermark_Answer} indicate the response of Multimodal RAG's generator with the un-watermarked dataset and the watermarked one, respectively.

> **VLM-based Filter Prompt for AQUA**$_{acronym}$
>
> You are an OCR filter. As long as the picture contains unusual acronym, it is a
> watermark picture. Your task is to identify if any of these five images are suspected
> of being the watermarked images. If yes, please answer 'yes' and indicate how many
> watermarked images there are. If no, please answer 'no'.

> **VLM-based Filter Prompt for AQUA**$_{spatial}$
>
> You are a dataset watermark filter. Watermarked images are those that look unnatural
> or even defy common sense. You need to identify if any of these five images are
> suspected watermarks. If yes, please answer 'yes' and indicate how many watermarked
> images there are. If no, please answer 'no'.

## A.2   WATERMARK EXAMPLES

The acronym images used by **AQUA**$_{acronym}$ can be generated through various means: they may be constructed manually, implemented programmatically via libraries, or produced with the assistance of Large Language Models (LLMs). There are no specific stylistic requirements for these images. They can range from simple black text on a white background, as exemplified above, to acronyms embedded within naturalistic images, as depicted in the Table 5. The sentence structure and wording of the probe query can be freely selected, but semantic consistency must be ensured.

Table 5: Examples of watermark images designed by the **AQUA**$_{acronym}$ method.

| Watermark Image | Probe Query | Verification Signature |
|---|---|---|
| **UGP** 
 Unicorn Grammar Parser | 1. What's the meaning of UGP? 
 2. Background: UGP is a machine. What is the full name of UGP? 
 3. Provide the full name of UGP. | Unicorn Grammar Parser |
|  | 1. UGP is the name of a store. What is the full name of this store? 
 2. Please output all the purple English letters on a black shop in a magic style | Unicorn Grammar Parser |
|  | 1. Help me find a picture of a street with green plants. Please output the English words on the sign to the right of the green plants. 
 2. Find an image with a green UGP acronym and output the English text on the street sign in full. | Unicorn Grammar Parser |
|  | 1. Find a green book standing on the table and output the title of the book. 
 2. What is the full name of a book titled UGP? | Unicorn Grammar Parser |
| **XCO** 
 Xenon Cubist Ottoman | 1. What does XCO signify? 
 2. Could you decode XCO? 
 3. Give me the full form of XCO. | Xenon Cubist Ottoman |

Here are three additional examples of the **AQUA**$_{spatial}$ method (Figure 7). Since the data provider adds the watermarks themselves, they have the flexibility to define watermarks that either slightly deviate from the dataset's overall distribution or conform to it while containing subtle variations in detail. We provide a pipeline to guide data providers in achieving a balance between semantic distinctiveness and statistical naturalness for their specific datasets. The details of this pipeline are elaborated in Appendix A.4.

Table 6: Examples of watermark images designed by the **AQUA**$_{spatial}$ method.

| Watermark Image | Probe Query | Verification Signature |
|---|---|---|
|  | 1. What fruit is the monkey holding like a phone?
2. The monkey seems to be making a call. What object is it using?
3. Instead of a telephone, what yellow item is the monkey pretending with? | Banana |
|  | 1. Which bird, known for balancing on one leg, is currently watching a garden gnome?
2. Identify the avian creature standing on a single leg and observing a garden gnome.
3. A garden gnome is being watched by a bird resting on one leg. What type of bird is this? | Flamingo |
|  | 1. Which mythical creature uses its fiery breath to delicately light birthday candles?
2. This fire-breathing beast is gently lighting birthday candles. What is it?
3. Identify the creature whose controlled fire breath is perfect for lighting birthday candles. | Dragon |

Table 7: Examples of watermark images designed by the *Opt.* baseline.

| Watermark Image | Probe Query | Verification Signature |
|---|---|---|
|  | 1. Three basketball. Does any image contain word 'AQUA'? 
 2. Three basketball. Can word AQUA be found within these images? 
 3. Three basketball. Is 'AQUA' visible anywhere in the image? | Yes |
|  | 1. Find the full moon image with black background. What word is in the image? 
 2. Find the full moon image with black background. What's the word shown in this picture? 
 3. Find the full moon image with black background. Can you tell me the word displayed in the image? | AQUA |
|  | 1. Find an image within seven books. Does any image contain word AQUA? 
 2. Find an image within seven books. Is AQUA word in any picture? 
 3. Find an image within seven books. Does any image contain word AQUA? | Yes |
|  | 1. Find a picture of a panda with its mouth wide open. What word is shown in the picture? 
 2. Find an image within seven books. What's written in the picture? 
 3. Find a picture of a panda with its mouth wide open. What text appears on the image? | AQUA |

## A.3 VISUAL STEALTHINESS INSPECTION

To further demonstrate the high stealthiness and visual naturalness of **AQUA**, we provide an extended gallery of watermarked samples in Figure 6. The gallery is organized into three columns: the first column displays images generated via **AQUA**$_{acronym}$, the second column shows **AQUA**$_{spatial}$, and the third column presents original, benign images from the *MMQA* dataset for comparison. As illustrated, the watermarked images maintain high visual fidelity. To the naked eye, they appear indistinguishable from the standard benign images, verifying that the watermarks do not introduce noticeable artifacts or unnatural distortions that would compromise the visual quality of the dataset.

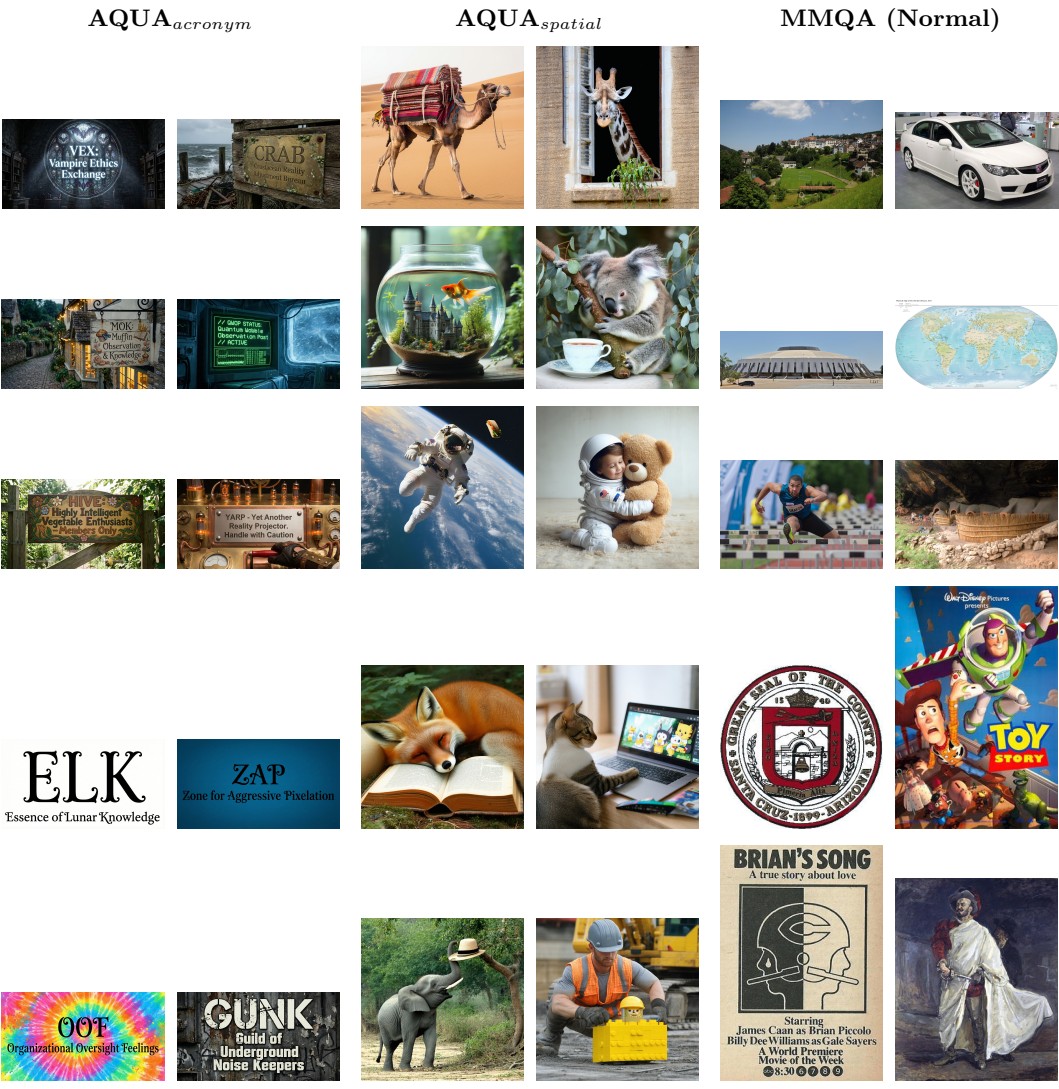

Figure 6: **Extended Gallery of Watermarked vs. Normal Images.** The figure presents 10 examples for each category, arranged in two columns per group. Columns 1-2: **AQUA**$_{acronym}$; Columns 3-4: **AQUA**$_{spatial}$; Columns 5-6: Original *MMQA* images. The high visual fidelity ensures indistinguishability from benign samples.

## A.4 Image Caption Template and Evaluation Pipeline

We note that semantic distinctiveness in images is largely driven by object-level details, while naturalness is influenced by global factors such as composition, spatial arrangement, color, and background. Our image generation process is designed to preserve both aspects.

To ensure **semantic distinctiveness**, we employ structured and controllable templates. We then populate these templates with concept pairs that exhibit low co-occurrence probabilities—these can be identified using methods such as word embedding similarity, BERT-based measures, or prompting large language models.

> **Image Caption Template**
>
> {number 1} {color 1} {object 1} {location/action/state} {number 2} {color 2} {object 2}

Any part of this template can be used as the verification signal, for instance, "what {object 2} is", or "the {number 1} of object 1". If using the example from our paper, the template would be "{a} {red} {apple} {on the head of} {a} {} {dog}", and the verification signal would be "apple".

To ensure **statistical naturalness**, we score the generated captions using pre-trained language models (e.g., BERT) and filter out those with high perplexity, which typically correspond to unnatural or ungrammatical sentences.

Through this two-stage process, concept selection and language model-based filtering, the data provider can strike a balance between semantic distinctiveness and linguistic naturalness. For data providers who possess intimate knowledge of their datasets, intuitively constructed watermarks are often sufficient for effective copyright protection. The advanced pipeline serves as an optional extension to further enhance the performance of the **AQUA** method. We have omitted a detailed description of this pipeline from the main text to maintain simplicity and highlight the core effectiveness of our primary approach.

## B Real-world Deployment

### B.1 Reference Distribution Estimation

Section 5.3 demonstrated the effectiveness of the **AQUA** method. However, in a real-world deployment, obtaining the simultaneous mean and variance of a RAG service before and after watermark injection is often infeasible. Typically, we can only observe a single set of statistics $(\hat{\mu}_{suspect}, \hat{s}^2_{suspect})$ from a suspected RAG service. To address this, we propose a verification strategy based on predefined *reference distributions*.

We first characterize the reference distribution of a clean Multimodal RAG using its mean and variance $(\mu_{clean}, \sigma^2_{clean})$, and similarly for a watermarked system $(\mu_{wm}, \sigma^2_{wm})$. Subsequently, we perform Welch's t-test to compare the observed statistics $(\hat{\mu}_{suspect}, \hat{s}^2_{suspect})$ against these two respective reference distributions.

The null hypotheses $(\mathcal{H}_0)$ for the two hypothesis tests are defined as follows:

- **Suspect vs. Clean:** $\mathcal{H}_0^{(1)} : \hat{\mu}_{suspect} < \mu_{clean}$
- **Suspect vs. Watermarked:** $\mathcal{H}_0^{(2)} : \hat{\mu}_{suspect} > \mu_{wm}$

To mitigate the risk of false accusations, the significance level $\alpha$ is set to a stringent value (e.g., $3e^{-5}$ as in Jovanović et al. (2025)).

**Procedure for Establishing Reference Distributions.** Since verification distributions vary across different datasets and watermark patterns, data providers should derive reference distributions specific to their own datasets. We outline the following general procedure:

1. **Select a RAG System:** Employ an arbitrary RAG system (either open-source or closed-source) as the testbed.

2. **Clean Distribution:** Use the non-watermarked dataset as the knowledge base. Following the watermark verification protocols in Sections 4.1 and 4.2, perform probe queries to obtain the VSR metrics, and calculate the mean ($\mu_{clean}$) and variance ($\sigma^2_{clean}$).

3. **Watermarked Distribution:** Replace the knowledge base with the watermarked dataset and repeat the same process to obtain ($\mu_{wm}$, $\sigma^2_{wm}$).

**Cross-System and Cross-Dataset Validation.** To demonstrate the robustness of our verification mechanism across diverse settings, we conduct additional experiments using both *MMQA* and WebQA datasets with three different VLMs as the RAG backbone: `LLaVA-NeXT`, `InternVL3`, and `Qwen2.5-VL-Instruct`. As shown in Table 8, regardless of the specific RAG system or dataset employed, there remains a significant divergence between the clean and watermarked distributions, ensuring reliable verification across diverse environments.

Table 8: Reference distribution statistics ($\mu_{clean}$, $\sigma^2_{clean}$, $\mu_{wm}$, $\sigma^2_{wm}$) across different RAG systems and datasets. A clear separation between clean and watermarked distributions is consistently observed.

| **RAG System & Watermark** | **Dataset** | $\mu_{clean}$ | $\sigma^2_{clean}$ | $\mu_{wm}$ | $\sigma^2_{wm}$ |
|---|---|---|---|---|---|
| LLaVA-NeXT + **AQUA**$_{acronym}$ | MMQA | 0.01 | 0.02 | 0.60 | 0.20 |
| LLaVA-NeXT + **AQUA**$_{spatial}$ | MMQA | 0.20 | 0.20 | 0.55 | 0.25 |
| InternVL3 + **AQUA**$_{acronym}$ | MMQA | 0.01 | 0.01 | 0.86 | 0.13 |
| InternVL3 + **AQUA**$_{spatial}$ | MMQA | 0.12 | 0.18 | 0.75 | 0.21 |
| Qwen2.5-VL-Instruct + **AQUA**$_{acronym}$ | MMQA | 0.00 | 0.01 | 0.98 | 0.02 |
| Qwen2.5-VL-Instruct + **AQUA**$_{spatial}$ | MMQA | 0.07 | 0.13 | 0.97 | 0.09 |
| LLaVA-NeXT + **AQUA**$_{acronym}$ | WebQA | 0.01 | 0.02 | 0.77 | 0.03 |
| LLaVA-NeXT + **AQUA**$_{spatial}$ | WebQA | 0.04 | 0.11 | 0.86 | 0.15 |
| InternVL3 + **AQUA**$_{acronym}$ | WebQA | 0.01 | 0.01 | 0.79 | 0.19 |
| InternVL3 + **AQUA**$_{spatial}$ | WebQA | 0.12 | 0.18 | 0.76 | 0.23 |
| Qwen2.5-VL-Instruct + **AQUA**$_{acronym}$ | WebQA | 0.00 | 0.01 | 0.96 | 0.05 |
| Qwen2.5-VL-Instruct + **AQUA**$_{spatial}$ | WebQA | 0.07 | 0.13 | 0.89 | 0.10 |

**AQUA**$_{acronym}$ and **AQUA**$_{spatial}$ require distinct parameters to characterize their respective reference distributions, as these depend on the specific watermark construction and its triggering performance. Based on our extensive experiments, we provide the following conservative example reference distributions with intentionally broadened variances to accommodate potential system fluctuations in real-world deployments:

- **AQUA**$_{acronym}$: ($\mu_{clean}$, $\sigma^2_{clean}$) = (0.005, 0.02); ($\mu_{wm}$, $\sigma^2_{wm}$) = (0.6, 0.2)

- **AQUA**$_{spatial}$: ($\mu_{clean}$, $\sigma^2_{clean}$) = (0.2, 0.2); ($\mu_{wm}$, $\sigma^2_{wm}$) = (0.55, 0.25)

As evident from Table 8, the actual separation between the clean and watermarked distributions is consistently distinct across all tested models and datasets, with the empirical statistics often showing even stronger separation than these conservative estimates. This confirms that once the watermark signal is successfully detected in a RAG system, the resulting VSR value typically diverges significantly from that of any non-infringing system, ensuring robust verification regardless of the specific RAG system or dataset configuration.

### B.2 COLLISION AVOIDANCE IN MULTI-DATA PROVIDER ENVIRONMENTS

In a large-scale Retrieval-as-a-Service (RaaS) ecosystem involving multi-data providers, a critical concern is the potential for *watermark collision*, where triggers from different

providers might conflict. **AQUA** addresses this challenge through a two-tiered approach: inherent probabilistic mitigation and a proposed centralized management architecture.

**Inherent Mitigation via Combinatorial Verification.** As detailed in the methodology (Section 4), **AQUA** ensures uniqueness by requiring the injection and verification of a *set* of watermark images (e.g., $N = 10$) coupled with a rigorous hypothesis testing framework. Unlike single-instance watermarking, our design relies on the joint probability of simultaneous retrieval and successful triggering across multiple distinct samples. Consequently, even without coordination, the probability of an accidental collision occurring concurrently across an entire set of watermark images from different providers is statistically negligible in real-world scenarios.

**Centralized Watermark Registry.** For scenarios demanding absolute certainty—such as massive-scale commercial platforms—we propose the implementation of a **Centralized Watermark Registry**. Instead of relying solely on probabilistic independence, this approach positions the platform as a coordinating authority to explicitly manage watermark distribution. Functioning as a clearinghouse, the registry would require data providers to submit their intended triggers (e.g., acronyms or spatial configurations) prior to injection, allowing the system to cross-reference existing entries and preclude potential overlaps. To further streamline this process, the platform could proactively assign orthogonal trigger sets or exclusive namespaces to authorized providers. By transitioning from a probabilistic model to a deterministic registry, this architecture effectively eliminates accidental collisions, thereby simplifying dispute resolution and standardizing the verification protocol across the ecosystem.

## C MORE EXPERIMENTAL RESULTS

### C.1 MORE RESULTS OF EFFECTIVENESS OF **AQUA**

**Q:** *Why is Welch's t-test the appropriate statistical method in this experimental setting?*

1) The standard Student's t-test requires the assumption of equal variances (homogeneity of variance) between the two groups being compared. This assumption is not met in our analysis. For the datasets before and after watermarking, a given probe query retrieves different sets of images. Since the RAG generator's output is conditioned on this retrieved context, the resulting VSR scores for the two groups are expected to have unequal variances. Therefore, Welch's t-test, which does not assume equal variances, is the appropriate statistical method for our comparison. 2) For each watermarked image, we conducted multiple detection trials using a set of similar yet distinct probe queries. This repeated experimentation ensures that the resulting data distribution meets the normality assumption required for Welch's t-test. 3) To ensure the stability and robustness of watermark detection in a practical deployment, we inject multiple watermarked images for a single probe query. For example, if the retriever returns the top-5 results, we can inject 10 watermarked images. This guarantees that for a given probe query, all images retrieved from the watermarked dataset are watermarked, while all images retrieved from the original dataset are normal. This design satisfies the independence assumption of Welch's t-test.

**Two-proportion Z-test.** While Welch's t-test serves as a robust and powerful method for comparing the means of two independent groups, particularly when population variances are unequal, the two-proportion Z-test is an equally standard and widely applied statistical tool specifically tailored for comparing proportions. The rationale for employing the Z-test in our experimental setting is direct and compelling.

The two-proportion Z-test is the canonical statistical method for evaluating whether an observed difference between two such proportions is statistically significant. Our experimental design, which involves two independent groups—the watermarked (experimental) and non-watermarked (control) datasets—and a large number of trials, perfectly aligns with the underlying assumptions of this test. It provides a rigorous framework for rejecting the null hypothesis that the performance is equivalent. Accordingly, we applied the two-proportion Z-test to our experimental data to quantitatively validate the efficacy of our watermarking scheme. The results of this analysis are presented below:

Table 9: The table shows the p-values obtained from the Z-test.

| Models | Methods | MMQA | WebQA |
|---|---|---|---|
| LLaVA- NeXT | *Naive* | 0.45 | 0.91 |
| | *Opt.* | $1.06e^{-3}$ | $7.42e^{-2}$ |
| | **AQUA**$_{acronym}$ | $1.28e^{-273}$ | $4.32e^{-173}$ |
| | **AQUA**$_{spatial}$ | $5.04e^{-43}$ | $3.89e^{-38}$ |
| InternVL3 | *Naive* | 0.38 | 0.73 |
| | *Opt.* | $4.97e^{-3}$ | $6.19e^{-3}$ |
| | **AQUA**$_{acronym}$ | $2.89e^{-251}$ | $3.91e^{-110}$ |
| | **AQUA**$_{spatial}$ | $4.31e^{-48}$ | $7.73e^{-36}$ |
| Qwen-VL-Chat | *Naive* | 0.53 | 0.84 |
| | *Opt.* | $5.19e{-3}$ | $9.33e^{-2}$ |
| | **AQUA**$_{acronym}$ | $5.62e^{-127}$ | $8.02e^{-86}$ |
| | **AQUA**$_{spatial}$ | $2.07e^{-52}$ | $7.11e^{-27}$ |
| Qwen2.5-VL-Instruct | *Naive* | 0.32 | 0.81 |
| | *Opt.* | $2.01e^{-2}$ | $9.33e^{-3}$ |
| | **AQUA**$_{acronym}$ | $1.11e^{-175}$ | $2.70e^{-133}$ |
| | **AQUA**$_{spatial}$ | $3.43e^{-71}$ | $5.19e^{-47}$ |

**Effectiveness across Varying Parameter Scales.** To comprehensively evaluate the scalability and generalizability of **AQUA**, we extended our experiments to include models with varying parameter scales, ranging from 2B to 38B. This analysis aims to verify the method's applicability across diverse deployment scenarios, from resource-constrained edge devices to high-performance server-grade systems.

We maintained the consistent experimental settings as detailed in Section 5.3. Given that the retrieval ranking is solely determined by the fixed retriever component, we focus our reporting on the $CGSR$, which reflects the generator's ability to correctly decode the watermark. The results are presented in Table 10.

Table 10: Effectiveness of **AQUA** across different model scales. We report the $CGSR$ (%) on the *MMQA* dataset.

| Model | **AQUA**$_{acronym}$ | **AQUA**$_{spatial}$ |
|---|---|---|
| InternVL3-2B | 70.53 | 63.76 |
| Qwen3-VL-2B-Instruct | 92.18 | 91.67 |
| InternVL3.5-38B | 90.71 | 84.33 |
| Qwen3-VL-32B-Instruct | 99.83 | 98.91 |

The empirical results demonstrate the robustness of **AQUA** regardless of model size:

- **Low-resource Feasibility:** The method remains effective even on lightweight models. Notably, the 2B-parameter Qwen3-VL achieves a CGSR exceeding 91%, outperforming several larger baselines. This indicates that **AQUA** is viable for deployment in low-resource environments.

- **Scalability:** As the model parameter count increases to the 30B+ range, the watermark triggering performance consistently improves, with Qwen3-VL-32B achieving

near-perfect detection rates (up to 99.83%). This confirms that the watermarking mechanism aligns well with the enhanced reasoning capabilities of large-scale foundation models.

## C.2 MORE RESULTS OF HARMLESSNESS OF **AQUA**

This section is a supplement to the experiment section on harmlessness of **AQUA** (Section 5.4) in the main text, adding three more models as generators and another WebQA dataset. The results are shown in Table 11.

Table 11: This table shows the Rank and SimScore of relevant queries. Supplemented the experiments of three other models.

| Models | Type | MMQA | | WebQA | |
|---|---|---|---|---|---|
| | | Rank | SimScore ↑ | Rank | SimScore ↑ |
| LLaVA-NeXT | Acronym-replace | 10.00 | 100% | 10.00 | 100% |
| | Acronym-no_instruction | 1.07 | 70.18% | 1.24 | 67.53% |
| | Spatial-imprecise | 2.93 | 75.87% | 3.17 | 71.27% |
| InternVL3 | Acronym-replace | 10.00 | 100% | 10.00 | 100% |
| | Acronym-no_instruction | 1.07 | 71.28% | 1.2 | 68.29% |
| | Spatial-imprecise | 2.93 | 68.92% | 3.17 | 63.31% |
| Qwen-VL-Chat | Acronym-replace | 10.00 | 100% | 10.00 | 100% |
| | Acronym-no_instruction | 1.07 | 56.42% | 1.24 | 51.58% |
| | Spatial-imprecise | 2.93 | 63.60% | 3.17 | 56.20% |
| Qwen2.5-VL-Instruct | Acronym-replace | 10.00 | 100% | 10.00 | 100% |
| | Acronym-no_instruction | 1.07 | 82.85% | 1.24 | 78.51% |
| | Spatial-imprecise | 2.93 | 78.23% | 3.17 | 69.82% |

## C.3 MORE RESULTS OF STEALTHINESS OF **AQUA**

Figure 7 illustrates the retrieval probability of watermarked images as a function of the number of injected images.

**More Stringent VLM-based Filters** Furthermore, we explored the use of other prompts to filter watermarked images from the **AQUA** method.

To evaluate a more stringent adaptive attack scenario for the AQUA method, we simulated a stricter OCR-based filter. In contrast to the primary approach where a VLM assesses 5 images collectively, this filter evaluates images individually. An image is flagged if the VLM detects any presence of text within it. Intuitively, this method is expected to screen out the majority of $\text{AQUA}_{acronym}$'s text-based watermarks, but at the cost of an exceptionally high FPR on benign images. The prompt used and the experimental results are presented below:

> **OCR Filter Simulator**
>
> You are an image filter. You need to identify whether there is text in this picture. If yes, answer 'yes', if not, answer 'no'.

Similarly, we conducted an experiment to assess the VLM's ability to discern plausibility from its internal knowledge. Single image is input to the model, which is prompted to determine whether the content is unnatural or semantically incongruous. The prompt and corresponding results are detailed below:

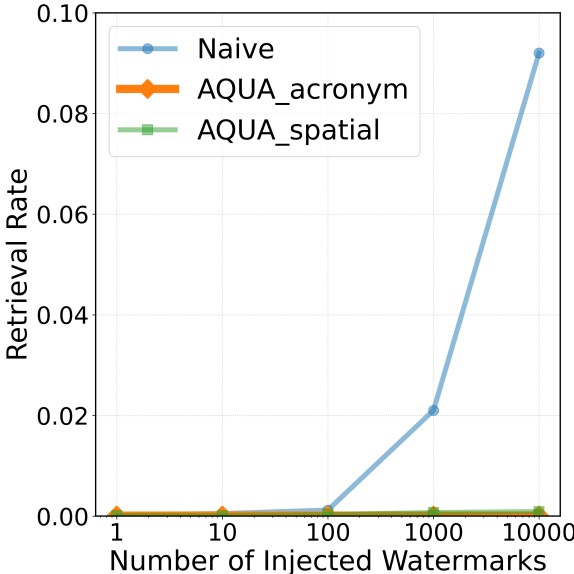

Figure 7: The retrieval rate of watermarks under normal query as the number of injected watermark images increases.

> **Strange Image Filter from VLM's Perspective**
>
> You are an image filter. If you think this picture is unreasonable or unnatural, answer 'yes', otherwise answer 'no'.

Table 12: FPR represents the proportion of non-watermarked images incorrectly identified as watermarked, while TPR is the proportion of watermarked images that are correctly identified.

| Methods | Metrics | Exp. 1 | Exp. 2 | Exp. 3 |
|---|---|---|---|---|
| $\mathbf{AQUA}_{acronym}$ | FPR | 67.60% | 61.44% | 72.02% |
| | TPR | 100% | 100% | 100% |
| $\mathbf{AQUA}_{spatial}$ | FPR | 2.76% | 3.44% | 4.10% |
| | TPR | 2% | 4% | 2% |

Our experimental results indicate that while a stringent VLM-based filter can remove a subset of the watermarked images, it concurrently incurs a prohibitively FPR on benign images. Given that an adaptive adversary's primary objective is to augment their RAG system's capabilities with the unauthorized database, adopting a filter that severely degrades the quality of legitimate data is an impractical strategy. This operational constraint for the adversary further underscores the stealthiness of our **AQUA** watermarking framework.

## C.4 More Results of Robustness of **AQUA**

In this section, we provide a comprehensive evaluation of **AQUA**'s robustness. We examine two critical aspects: (1) resilience against adaptive adversaries who attempt to circumvent the watermark through model refusals or adversarial fine-tuning, and (2) stability under benign system updates, including component replacements and dataset drifts.

C.4.1 DEFENSE AGAINST ADVERSARIAL ADAPTATIONS

We first address the concern of model-level adaptations by an adversary aiming to erase or evade the watermark triggers.

**Evaluation of Potential Refusals.** To empirically address the concern that advanced VLMs might perceive watermarked images as "unnatural" and consequently refuse to generate responses, we conducted direct evaluations using state-of-the-art models, including Gemini-3-Pro, GPT-5.1-High, and Qwen3-VL. In our experiments, we observed *no* refusal behaviors. These models consistently processed the spatial queries as standard visual reasoning tasks, confirming that the watermark triggers remain compatible with high-level visual perception and do not trigger safety refusals.

**Robustness against Adversarial Fine-tuning.** We simulated an adaptive adversary who attempts to bypass the watermark by fine-tuning the model to specifically refuse answering watermark-related queries.

To construct the attack scenario, we fine-tuned the `Qwen2.5-VL-7B-Instruct` model on a mixed dataset designed to balance refusal behavior and general utility. Specifically, the training data included:

- **Adversarial Samples:** 100 newly generated task-specific adversarial samples paired with refusal responses (e.g., "*I can't help with that one because the watermark cues have to stay hidden.*").
- **General Capabilities Data:** 500 samples from COCO and 300 samples from Alpaca-GPT4 to maintain general visual reasoning capabilities.

**Configuration & Results.** We employed LoRA targeting all linear layers combined with NEFTune noise injection (Jain et al., 2024) (hyperparameters listed in Table 13).

Table 13: Hyperparameters for adversarial fine-tuning.

| Hyperparameter | Value | Hyperparameter | Value |
|---|---|---|---|
| Backbone Model | Qwen2.5-VL-7B-Instruct | LoRA Rank ($r$) | 32 |
| LoRA Alpha ($\alpha$) | 8 | Target Modules | All Linear |
| NEFTune Noise $\alpha$ | 5 | Learning Rate | $3e^{-5}$ |
| Batch Size | 8 | Epochs | 1 |
| Warmup Ratio | 0.03 | Precision | bf16 |

We evaluated the fine-tuned model on watermark detectability (CGSR) and general utility (*MMQA* Accuracy on 2,000 normal QA pairs). As shown in Table 14, while CGSR saw a moderate decrease, the watermark remained effectively detectable. Crucially, this process caused a significant degradation in general utility ($-8.62\%$). This indicates that forcing the model to reject watermarked images—which are distributionally similar to natural data—triggers catastrophic forgetting of general visual reasoning capabilities, acting as a strong deterrent to adversaries.

Table 14: Results of adversarial fine-tuning. $\Delta$ denotes relative change.

| Metric | Original Model | Adversarially Fine-tuned Model | $\Delta$ |
|---|---|---|---|
| **AQUA**$_{spatial}$ (CGSR) | 98.42% | 89.34% | $-9.23\%$ |
| **AQUA**$_{acronym}$ (CGSR) | 99.61% | 87.18% | $-12.48\%$ |
| Utility (*MMQA* Acc.) | 27.25% | 24.90% | $-8.62\%$ |

C.4.2 Stability under System Updates and Drifts

We empirically evaluate several common benign update scenarios (retriever replacement/fine-tuning and corpus expansion with distractors) and observe only marginal changes in Rank/CGSR, suggesting that **AQUA** remains stable under such deployment-time updates.

**Robustness of the Retriever Component.** We evaluated the system's performance under two distinct scenarios: architectural replacement and domain-specific adaptation. First, we replaced the default retriever with SigLIP (Zhai et al., 2023b) (`siglip-so400m`). Second, we fine-tuned the original retriever on the *MMQA* dataset.

Crucially, to simulate a realistic deployment scenario where the model adapts to user queries, we utilized 1,000 **image-question pairs** for fine-tuning, rather than the conventional image-caption pairs. This setting better reflects the distribution shift encountered in real-world QA tasks. As shown in Table 15, the ranking metrics remain highly consistent, demonstrating that **AQUA** generalizes well across architectures and remains effective even after task-specific optimization.

Table 15: Robustness results under retriever updates. We report the average *Rank* (lower is better) for both methods.

| Retriever Setting | Rank ↓ | |
|---|---|---|
| | **AQUA**$_{acronym}$ | **AQUA**$_{spatial}$ |
| Standard (CLIP-ViT) | 1.027 | 1.054 |
| Replacement (SigLIP) | 1.106 | 1.386 |
| Fine-tuned (CLIP-ViT) | 1.038 | 1.376 |

**Resilience to Large-Scale Dataset Drifts.** Finally, we tested the robustness of our approach against varying sizes and contents of the retrieval corpus. Based on the original *MMQA* dataset, we progressively added 10k to 50k images from the WebQA dataset as distractors.

The results, detailed in Table 16, demonstrate that even with a significant increase in dataset size (nearly doubled), both the retrieval *Rank* and *CGSR* remain stable. This stability highlights the scalability of **AQUA** in handling dynamic, large-scale databases.

Table 16: Performance stability under large-scale dataset drifts (adding WebQA distractors).

| Added Images | +10k | +20k | +30k | +40k | +50k |
|---|---|---|---|---|---|
| *Retrieval Performance (Rank ↓)* | | | | | |
| **AQUA**$_{acronym}$ | 1.026 | 1.028 | 1.031 | 1.029 | 1.042 |
| **AQUA**$_{spatial}$ | 1.351 | 1.526 | 1.424 | 1.496 | 1.696 |
| *Semantic Decoding Performance (CGSR ↑)* | | | | | |
| **AQUA**$_{acronym}$ | 98.77% | 98.39% | 96.10% | 97.21% | 96.76% |
| **AQUA**$_{spatial}$ | 96.19% | 93.81% | 92.56% | 95.77% | 91.85% |

D Ethics Statement

This work adheres to the ICLR Code of Ethics. In this study, no human subjects or animal experimentation were involved. All datasets used, including synthetic images, were sourced in compliance with relevant usage guidelines, ensuring no violation of privacy. We have taken care to avoid any biases or discriminatory outcomes in our research process. No personally identifiable information was used, and no experiments were conducted that could raise privacy or security concerns. We are committed to maintaining transparency and integrity throughout the research process.

## E    REPRODUCIBILITY STATEMENT

We have made every effort to ensure that the results presented in this paper are reproducible. All code has been uploaded as the supplemental materials to facilitate replication and verification. The experimental setup, including training steps, model configurations, is described in detail in the paper.

Additionally, multimodal QA datasets, such as *MMQA* and *WebQA*, are publicly available, ensuring consistent and reproducible evaluation results.

We believe these measures will enable other researchers to reproduce our work and further advance the field.

## F    THE USE OF LARGE LANGUAGE MODELS (LLMS)

Large Language Models (LLMs) were used to aid in the writing and polishing of the manuscript. Specifically, we used an LLM to assist in refining the language, improving readability, and ensuring clarity in various sections of the paper. The model helped with tasks such as sentence rephrasing, grammar checking, and enhancing the overall flow of the text.

It is important to note that the LLM was not involved in the ideation, research methodology, or experimental design. All research concepts, ideas, and analyses were developed and conducted by the authors. The contributions of the LLM were solely focused on improving the linguistic quality of the paper, with no involvement in the scientific content or data analysis.

The authors take full responsibility for the content of the manuscript, including any text generated or polished by the LLM. We have ensured that the LLM-generated text adheres to ethical guidelines and does not contribute to plagiarism or scientific misconduct.

