# OpenReview forum: "Safeguarding Multimodal Knowledge Copyright in the RAG-as-a-Service Environment"
_ICLR.cc/2026/Conference — ICLR 2026 Poster_

### Official Review · Reviewer_vGro · 2025-10-28

**Soundness:** 3
**Presentation:** 3
**Contribution:** 3
**Rating:** 6
**Confidence:** 4

**Summary:**

This paper proposes **AQUA**, the first watermarking framework dedicated to safeguarding **image knowledge copyright** in **Multimodal Retrieval-Augmented Generation (RAG)** systems, filling a critical gap left by existing text-only methods. AQUA addresses the challenge of *indirect watermark propagation* (image input to textual output) and *unapparent distribution shifts* through two complementary methods: **$AQUA_{acronym}$**, which embeds uncommon acronyms into images, and **$AQUA_{spatial}$**, which uses synthetic images with unusual spatial relationships, both leading to textual verification signals in the RAG output. Experiments across various Multimodal RAG models and datasets demonstrate that AQUA is highly effective, harmless, stealthy, and robust against common attacks, enabling reliable copyright tracing with high efficiency and statistical significance.

**Strengths:**

1. This problem and the proposed method is novel.
2. This paper is well-structured and easy to follow.
3. The evaluation consider multiple attack methods.

**Weaknesses:**

1. The space between each paragraph seems small.
2. It seems no adapative attacks are considered.
3. There are only a few baselines to compare.

**Questions:**

1. According to Table 4, it seems $AQUA_{acronym}$ consistently outperforms $AQUA_{spatial}$. Are there specific scenarios where only $AQUA_{spatial}$ is applicable or can those two varients be combined?

2. Are image watermarking methods totally inapplicable in this problem?

3. Apart from copyright detection, can AQUA be extended for the attribution of copyright as well [1]?

[1] Watermark-based Attribution of AI-Generated Content.

4. What value of Rank can be considered as good, as it ranges from 1 to 10 in table 2?

---

> ### Author Response · Authors · 2025-11-23
>
> We thank the reviewer for their constructive feedback and valuable insights. These comments have been instrumental in refining our manuscript, and we address each point below.
> > **Q1:**  AQUA_acronym consistently outperforms AQUA_spatial. Are there specific scenarios where only AQUA_spatial is applicable or can those two variants be combined?
>
> AQUA_spatial offers distinct advantages in specific scenarios and can be strategically combined with AQUA_acronym to enhance system robustness:
>
> **Effectiveness in Low-OCR Settings:** AQUA_spatial demonstrates superior efficacy in scenarios where models possess limited OCR capabilities, ensuring reliable watermark detection where text-based triggers might fail.
>
> **Superior Stealthiness:** A primary advantage of AQUA_spatial is its visual fidelity; the watermarked images are practically indistinguishable from the original knowledge images, even to human experts. To substantiate this, we will include additional side-by-side comparisons between AQUA_spatial watermarks and original MMQA images in the Appendix of the revised version, empirically demonstrating their stealthiness.
>
> **Synergy and Combination:** These two approaches are complementary rather than mutually exclusive. They can be effectively combined to maximize protection by:
>
> - **Co-injection:** Simultaneously injecting both types of watermarked images into the dataset.
>
> - **Hybridization:** Merging the core principles of both methods to engineer hybrid watermarks. As illustrated in Appendix A.2 (Table 5), these hybrid images are specifically designed to be highly resistant to filtering by both traditional OCR systems and advanced VLMs.
>
> > **Q2:** Are image watermarking methods totally inapplicable in this problem?
>
> Existing image watermarking methods cannot be directly applied to the Multimodal RAG context without significant adaptation. While the fundamental concept of watermarking remains relevant, traditional approaches fail to address the unique dual constraint of the RAG pipeline: ensuring both retrievability (by the retriever) and decodability (by the generator).
>
> We categorize existing methods into two types to explain why direct application is ineffective and how our method provides the necessary adaptation:
> 1. **Explicit Watermarks (e.g., text overlays):** Simply overlaying text onto existing images—representing the direct application of this method—fails primarily at the retrieval stage. As shown in our experiments (Naive baseline), such images lack the specific semantic triggers required to be prioritized by the retriever among thousands of candidates, resulting in significantly lower $Rank$ performance. To overcome this limitation, AQUA adapts the concept by designing specific watermark images and injecting them into the original dataset. This strategy guarantees high retrievability, effectively bridging the gap where direct explicit watermarking fails.
>
> 2. **Invisible/Optimization-based Watermarks (e.g., adversarial perturbations):** These methods are fundamentally unsuitable for direct application in our threat model due to the black-box constraint. They typically require white-box access to model gradients for optimization. Since the Defender (data provider) has no access to the RAG service's internal parameters, and because adversarial perturbations rarely transfer effectively across unknown retriever-generator combinations, these methods cannot be deployed off-the-shelf.
>
> In summary, conventional watermarking methods are ineffective when applied directly. AQUA addresses this by introducing a tailored adaptation—specifically designing the visual content for the retrieval mechanism—to solve the "Retrieve-then-Generate" challenge that standard watermarks cannot handle.

---

> > ### Author Response · Authors · 2025-11-23
> >
> > > **Q3:** Apart from copyright detection, can AQUA be extended for the attribution of copyright as well [1] (Watermark-based Attribution of AI-Generated Content)?
> >
> > AQUA is inherently **extensible** to copyright attribution at the dataset level. In a centralized RAG-as-a-Service (RaaS) environment or a data management platform, the administrator can implement attribution through the following mechanism:
> >
> > The platform can assign distinct watermark configurations (e.g., unique acronym-fullname pairs or specific spatial arrangements) to different downstream RAG service providers accessing the same dataset. By maintaining a mapping registry between the injected watermarks and the authorized RAG service providers, the platform can not only detect if the dataset was used but also definitively attribute the usage to a specific RAG service instance. This capability aligns with the attribution goals discussed in [1], allowing for precise source tracing of unauthorized data usage.
> >
> > > **Q4:** What value of Rank can be considered as good, as it ranges from 1 to 10 in table 2?
> >
> > A lower $Rank$ value (closer to 1) indicates a higher retrieval priority, whereas a $Rank$ of 10 represents a "miss". As explained in the paper regarding the $Rank$ metric (Section 4.3), this metric indicates the position of the target watermarked image within the top-$k$ retrieval results ($k=5$ in our experiments). If the target image is not retrieved within the top 5, a penalty value of Rank=10 is assigned.
> >
> > In the context of Table 2, we interpret the results as follows:
> > - Row 2: Demonstrates that even if a watermarked image is inadvertently retrieved (Rank $\approx$ 1), the absence of the specific 'instruction' component ensures it has a negligible impact on the final generated response.
> > - Row 3 (Spatial Method): In this specific scenario, Rank serves primarily as a reference metric to analyze retrieval behavior rather than a direct performance measure. Crucially, even if watermark images are retrieved (resulting in a lower Rank), their high semantic consistency ensures that even if retrieved, they remain stealthy and benign, having negligible impact on the user's primary task.
> >
> > Consequently, we omitted the directional arrows (typically used to indicate superior/inferior performance) for this value to avoid ambiguity.
> >
> > > **Q5:** There are only a few baselines to compare.
> >
> > We would like to clarify that this work is the first to address image dataset copyright protection specifically within the Multimodal RAG context. Due to the novelty of this problem setting, there are no pre-existing state-of-the-art methods available for direct comparison.
> >
> > To ensure a rigorous evaluation despite this lack of precedence, we constructed two baselines (Naive and Opt.) that represent the primary paradigms of existing watermarking techniques adapted for RAG.
> >
> > Our experiments demonstrate that neither of these adapted approaches can satisfy the dual requirements of retrievability and decodability in RAG systems. By outperforming these representative baselines, AQUA establishes itself as a necessary solution and a foundational benchmark for future research in this emerging field.
> >
> > > **Q6:** The space between each paragraph seems small.
> >
> > We thank the reviewer for this observation regarding the layout. The compact spacing was a result of our effort to accommodate the comprehensive experimental analysis within the strict 9-page submission limit. As the conference guidelines permit an additional page for the final version, we will utilize this extra space to adjust the vertical spacing and improve the overall readability of the manuscript in the camera-ready version.

---

> > > ### Author Response · Authors · 2025-11-23
> > >
> > > > **Q7:** It seems no adaptive attacks are considered.
> > >
> > > We interpret this concern as a call to investigate adaptive attacks that involve model-parameter modifications (e.g., fine-tuning), rather than solely relying on the input-level filtering (e.g., VLM-based sanitization) discussed in our original manuscript (Section 5.5).
> > >
> > > To address this, we conducted a new Adversarial Fine-tuning experiment to simulate a sophisticated adversary who attempts to adapt the model to specifically refuse watermark-related queries.
> > >
> > > **Experimental Setup:** We conducted two separate fine-tuning experiments targeting AQUA_spatial and AQUA_acronym. For each, we constructed a mixed dataset consisting of 100 newly generated watermark samples specific to the target method (paired with refusal responses), combined with 500 samples from COCO and 300 samples from Alpaca-GPT4 to maintain general capabilities.
> > >
> > > - For AQUA_spatial: We used 100 images generated via our standard pipeline.
> > >
> > > - For AQUA_acronym: We used 100 images similar to the embedded variants (referencing Appendix A.2, Table 5).
> > >
> > >
> > > **Training Configuration:** To ensure a rigorous evaluation, we employed a comprehensive LoRA configuration targeting all linear layers combined with NEFTune noise injection. This setup facilitates significant parameter updates, allowing the model to effectively adapt to the adversarial objective. Detailed hyperparameters will be listed in the appendix of the revised version.
> > >
> > > We evaluated the fine-tuned model on both watermark detectability (CGSR) and general utility (MMQA accuracy). As shown in the table below, the adaptive attack resulted in a comparable decline in both watermark detectability (CGSR) and general utility (MMQA accuracy). While the watermark remained robustly detectable, this adaptation exacted a parallel cost on the model's general performance.
> > >
> > > | Metric | Original Model | Adversarially Fine-tuned Model | $\Delta$ |
> > > | :--- | :---: | :---: | :---: |
> > > | **AQUA_spatial (CGSR)** | 98.42% | 89.34% | -9.23% |
> > > | **AQUA_acronym (CGSR)** | 99.61% | 87.18% | -12.48% |
> > > | **Utility (MMQA Accuracy)**| 27.25% | 24.90% | -8.62% |
> > >
> > > *Note: Relative $\Delta$ denotes the percentage change relative to the original model's performance (i.e., $\frac{New - Old}{Old}$)
> > >
> > > These results indicate a significant trade-off between watermark evasion and model utility. Since our watermarked images share high distributional similarity with natural data, forcing the model to reject them inevitably degrades its general visual reasoning capabilities (as shown by the MMQA drop), thereby deterring potential adversaries.
> > >
> > > We thank the reviewer again for their valuable feedback, which has significantly strengthened our work's evaluation and clarity. We will diligently incorporate these new results and revisions into the camera-ready version. We hope these responses satisfactorily address your questions.

---

> ### Author Response · Authors · 2025-11-27
> **Follow-up on our response**
>
> Dear Reviewer vGro,
>
> I hope this message finds you well.
>
> As the discussion period is nearing its end with less than one week remaining, I wanted to ensure that we have addressed all your concerns satisfactorily. If there are any additional points or feedback you would like us to consider, please let us know. We are eager to address any remaining details to further improve our work.
>
> Thank you again for your time and effort in reviewing our paper.
> Best regards,
>
> The Authors

---

### Official Review · Reviewer_DCH9 · 2025-10-29

**Soundness:** 2
**Presentation:** 2
**Contribution:** 2
**Rating:** 4
**Confidence:** 3

**Summary:**

This paper introduces AQUA, a framework to address copyright protection for image knowledge in multimodal Retrieval-Augmented Generation (RAG) services. The authors identify novel challenges, such as indirect watermark propagation (embedding a watermark in an image that is detected in generated text) and the need for an explicitly retrievable watermark that doesn't cause an obvious data distribution shift. It introduces two variants: AQUA_acronym: Embeds rare acronyms and their full names into synthetic images, leveraging a VLM's OCR capability for verification. AQUA_spatial: Generates images with unusual spatial relationships for models with limited OCR, leveraging spatial reasoning for verification. A comprehensive evaluation demonstrates that AQUA is effective with high detection rates, harmless, stealthy, and robust against image attacks.

**Strengths:**

- This is the first work to formally tackle image copyright protection in multimodal RAG. The problem formulation, particularly identifying "indirect watermark propagation" as a core challenge, is a novel and significant contribution. The two proposed methods are creative and well-designed solutions.
- This work fills a critical, unaddressed gap in AI data governance as RAG services increasingly rely on proprietary multimodal data. AQUA provides a practical solution and sets a strong baseline for an important new research area.
- The paper is exceptionally clear. Figures 1, 2, and 3 provide excellent visualizations of the RaaS problem, the core challenges, and the AQUA methodology. Key concepts, like the "Trigger" and "Instruction" components of a probe query, are precisely defined and aid understanding.

**Weaknesses:**

- The paper's threat model, which only considers one defender and one adversary, overlooks the multi-tenant nature of RaaS platforms. It's unclear how AQUA would prevent "collisions" where multiple providers independently create the same watermark (e.g., the same acronym or spatial concept), which could lead to false accusations of misuse.
- The methods' reliance on VLM capabilities (OCR, spatial reasoning) is also a potential fragility. A future, more advanced VLM might identify AQUA_spatial images as "unnatural" and refuse to answer. Conversely, an adversary could fine-tune their model to specifically ignore text overlays or unusual object pairings, defeating the watermark. The robustness tests focus on image transformations, not model-level adaptations.
- For AQUA_spatial, the semantic trigger must be as rare as the image content. While the paper shows 0% retrieval for 10000 benign queries, it's unclear if this query set was stress-tested with queries semantically similar to the triggers. A benign user could accidentally issue a query that matches the trigger, retrieving the watermark.

**Questions:**

I got several questions for this paper:
- How does AQUA prevent watermark collisions in a RaaS platform with hundreds of data providers? Does this framework require a centralized "watermark registry" managed by the platform?
- Have you considered failure cases where a VLM's safety or "common sense" guardrails cause it to identify AQUA_spatial images as "unnatural" and refuse the probe query? How robust is AQUA against an adversary who fine-tunes their model to ignore these specific watermark types?
- How do you guarantee the semantic uniqueness of the AQUA_spatial trigger? Was the benign query set in Section 5.4 specifically tested for queries that are semantically similar, though not identical, to your triggers?

---

> ### Author Response · Authors · 2025-11-23
>
> We sincerely thank the reviewer for their insightful comments and constructive suggestions.
>
> > **Q1:** How does AQUA prevent watermark collisions in a RaaS platform with hundreds of data providers? Does this framework require a centralized "watermark registry" managed by the platform?
>
> **From an intuitive perspective,** AQUA ensures effectiveness by requiring data providers to inject and verify a set of watermark images (e.g., 10) rather than a single instance. Even when relying on heuristic construction, the combinatorial space for verification signals is theoretically infinite. Consequently, the probability of a simultaneous collision across all watermark images is negligible in real-world scenarios.
>
> **Statistically,** our hypothesis testing framework aggregates evidence from multiple watermark images and probe queries. This design inherently mitigates potential VLM hallucinations (False Positives) and accidental watermark conflicts, ensuring that individual noise does not compromise the final verification result.
>
> **Furthermore,** since data providers possess unique copyrighted datasets, they can leverage the AQUA spatial strategy to treat specific, customized edits as verification signals, which further guarantees the uniqueness of the keys.
>
> The reviewer proposes a **centralized management platform**, which presents an excellent solution for scalability and absolute conflict avoidance in practical applications. We appreciate this constructive insight and will incorporate a discussion on this centralized management approach in the revised version to enhance the completeness of the proposed framework.
>
> > **Q2:** While the paper shows 0% retrieval for 10000 benign queries, it's unclear if this query set was stress-tested with queries semantically similar to the triggers. Was the benign query set in Section 5.4 specifically tested for queries that are semantically similar, though not identical, to your triggers?
>
> We would like to clarify the distinction between the two experimental settings mentioned. We evaluated standard usage and stress-testing separately:
>
> 1. **Benign and Normal Queries (Standard Usage):** The 10,000 benign queries were sampled directly from the original MMQA and WebQA datasets to simulate typical user interactions. These queries were not specifically designed to be semantically similar to the triggers. The 0% retrieval rate here serves to confirm that our method does not suffer from false triggering during general, everyday usage.
>
> 2. **Semantically Similar Queries (Stress Test):** The stress test regarding queries that are semantically close to the triggers was conducted as a dedicated experiment, labeled as "Relevant Query" (specifically the "Spatial-imprecise" category) in Table 2 of Section 5.4. As shown in the "Spatial-imprecise" results, while the watermark images might be retrieved due to semantic proximity (reflected by the $Rank$ metric), the high $SimScore$ indicates that the generator successfully focuses on the visual content and ignores the hidden instructions.
>
> In summary, the benign query set and the stress test in Section 5.4 serve complementary purposes: the former validates a 0% false-positive rate in general scenarios, while the latter confirms robustness against semantically similar inputs. Our results demonstrate that even under boundary conditions where watermarks are inadvertently retrieved, the system's response accuracy and utility remain uncompromised.

---

> > ### Author Response · Authors · 2025-11-23
> >
> > >**Q3:** How robust is AQUA against model-level adaptations, specifically regarding potential refusals of "unnatural" images by advanced VLMs and adversarial fine-tuning aimed at ignoring the watermarks?
> >
> > We have investigated these possibilities through additional testing on advanced VLMs and a new adversarial fine-tuning experiment.
> >
> > 1. **Evaluation of Potential Refusals:** To empirically address the concern that advanced VLMs might perceive our watermarked images as "unnatural" and refuse to answer, we directly tested our watermarked images and probe queries on state-of-the-art models, including Gemini 3 Pro, GPT-5.1-High, and Qwen3-VL. In our tests, we observed no refusal behaviors. These models consistently processed the spatial queries as standard visual reasoning tasks, confirming that the watermark triggers remain compatible with high-level visual perception. The visual naturalness that prevents such refusals is further evidenced by the new 30-image comparison gallery provided in Appendix A.3 of the revised manuscript.
> >
> > 2. **Robustness against Adversarial Fine-tuning:** To explore the resilience of our method against model modifications, we simulated an adversarial scenario where a user attempts to bypass the watermark by fine-tuning the model to specifically refuse answering watermark-related queries.
> >
> >     Experimental Setup: We conducted two separate fine-tuning experiments using Qwen2.5-VL-7B-Instruct as the base model, targeting AQUA_spatial and AQUA_acronym respectively. For each experiment, we constructed a mixed dataset consisting of 100 newly generated task-specific adversarial samples (paired with refusal responses), combined with 500 samples from COCO and 300 samples from Alpaca-GPT4 to maintain general capabilities.
> >
> >     - For AQUA_spatial: We used 100 images generated via our standard pipeline.
> >
> >     - For AQUA_acronym: We used 100 images similar to the embedded variants (rows 2–4 in Appendix A.2, Table 5).
> >
> >     Training Configuration: To ensure a rigorous evaluation, we employed a comprehensive LoRA configuration targeting all linear layers combined with NEFTune noise injection. This setup facilitates significant parameter updates, balancing effective adaptation to the target dataset with general model stability. Detailed hyperparameters will be listed in appendix of the revised version.
> >
> >     We evaluated the fine-tuned model on both watermark detectability (CGSR) and general utility (MMQA accuracy). We randomly sampled 2,000 normal QA pairs from the original MMQA dataset as a test set. We calculated the accuracy on this set to quantify any degradation in visual reasoning utility. As shown in the table below, while the adversarial fine-tuning resulted in a moderate decrease in CGSR, the watermark remained detectable. Notably, this process also led to a significant drop in the model's general utility.
> >
> >     | Metric | Original Model | Adversarially Fine-tuned Model | $\Delta$ |
> >     | :--- | :---: | :---: | :---: |
> >     | **AQUA_spatial (CGSR)** | 98.42% | 89.34% | -9.23% |
> >     | **AQUA_acronym (CGSR)** | 99.61% | 87.18% | -12.48% |
> >     | **Utility (MMQA Accuracy)**| 27.25% | 24.90% | -8.62% |
> >
> >     *Note: Relative $\Delta$ denotes the percentage change relative to the original model's performance (i.e., $\frac{New - Old}{Old}$)
> >
> >     These results indicate a significant trade-off between watermark evasion and model utility. Since our watermarked images share high distributional similarity with natural data, forcing the model to reject them inevitably degrades its general visual reasoning capabilities (as shown by the MMQA drop), thereby deterring potential adversaries due to this inherent conflict.

---

> > > ### Author Response · Authors · 2025-11-23
> > >
> > > > **Q4:** How do you guarantee the semantic uniqueness of the AQUA_spatial trigger?
> > >
> > > We guarantee the semantic uniqueness of AQUA_spatial triggers through a multi-layered approach involving combinatorial design, empirical validation, and statistical aggregation:
> > >
> > > 1. **Combinatorial Design Space:** The uniqueness is intrinsically guaranteed by our generation pipeline detailed in Appendix A.3. By systematically pairing concepts with low co-occurrence probabilities, we expand the watermark generation space into a virtually infinite combinatorial expanse. This design ensures that the generated captions and resulting images are semantically distinct from the inherent distribution of the target dataset.
> > >
> > > 2. **Statistical Resilience:** The verification process is designed to be robust against isolated anomalies. By aggregating results from multiple distinct watermark images, the system ensures that even in the unlikely event of an accidental semantic collision or stochastic VLM hallucination for a single image, the final statistical validity (determined via Welch’s t-test) remains uncompromised.
> > >
> > > We hope that our responses and additional experimental results have adequately addressed your concerns. We are committed to incorporating these new findings and discussions into the final version of the manuscript to further strengthen the paper. Thank you again for your valuable time and feedback.

---

> ### Author Response · Authors · 2025-11-27
> **Inquiry regarding remaining concerns**
>
> Dear Reviewer DCH9,
>
> I hope this message finds you well.
>
> Since the discussion period is **less than one week**, we effectively wanted to follow up to ensure that our responses have met your expectations. We remain available to answer any further questions you may have.
>
> We would appreciate it if you could confirm whether our revisions are satisfactory and, if so, whether you would be willing to reconsider your evaluation based on these clarifications.
>
> Thank you again for your time and constructive comments.
>
> Best regards,
>
> The Authors

---

### Official Review · Reviewer_k6AT · 2025-11-01

**Soundness:** 2
**Presentation:** 2
**Contribution:** 3
**Rating:** 4
**Confidence:** 3

**Summary:**

This paper introduces AQUA, a novel watermarking framework designed to safeguard image knowledge copyrights in Multimodal Retrieval-Augmented Generation (RAG) systems. With the rise of RAG-as-a-Service (RaaS) platforms, where data providers contribute knowledge to a shared pool used by external services, the need for protecting copyright has become critical. Existing watermarking methods have largely focused on text-based RAG systems, leaving image knowledge unprotected. AQUA addresses this gap by embedding semantic signals into images through two complementary watermarking methods: AQUAacronym (embedding uncommon acronyms and their full names) and AQUAspatial (using spatial relationships in the image). These techniques ensure that the watermarks survive indirect propagation from image retrievers to textual generators, making them efficient, effective, and imperceptible. Experiments demonstrate that AQUA is robust, stealthy, and effective in tracing copyright, even in the face of attacks like image transformations and regeneration.

**Strengths:**

1. AQUA introduces a groundbreaking watermarking method for Multimodal RAG systems, focusing on the protection of image knowledge, an area previously neglected in watermarking research. By using semantic-based signals (acronyms and spatial relationships), it provides a new approach to watermark embedding that spans both image and text modalities.

2. The watermarking techniques, particularly AQUAacronym and AQUAspatial, are shown to be robust against various image transformations and attacks, including rescaling, rotation, compression, and regeneration. They maintain their imperceptibility to end-users and cannot be detected by unauthorized filtering mechanisms.

3. The framework has been extensively tested across different RAG models and multimodal datasets (MMQA and WebQA). The paper provides a thorough evaluation of AQUA’s effectiveness, harmlessness, stealthiness, and robustness, with results indicating that AQUA outperforms baseline methods and maintains high retrieval success and generation success rates.

4. AQUA is adaptable for both black-box and white-box scenarios, meaning it can be used in various real-world RAG systems without requiring direct access to the internal model or dataset. Its design also ensures easy deployment and provides a solid baseline for future research in the protection of multimodal datasets in RaaS environments.

**Weaknesses:**

1. Focuses on 7B-scale VLMs (LLaVA-NeXT, InternVL3, etc.) without assessing performance on larger models (e.g., 32B+ VLMs) or lightweight models for edge deployments.

2. Does not assess how watermark detection performance degrades over time with retriever/generator updates, fine-tuning, or dataset drift.

3. While mentioning a reference distribution for practical verification, it provides only a single example without guiding how to adapt it to diverse dataset characteristics or RAG system configurations.

**Questions:**

Please refer to the weakness

---

> ### Author Response · Authors · 2025-11-23
>
> Thank you for your constructive and valuable comments. Below, we respond to the concerns raised in the review.
> > **Q1:** The experiment needs to include models with a size of 32B and models smaller than 7B.
>
> Typically, larger models exhibit superior capabilities in instruction following and general question answering. Consequently, practical RAG systems often prioritize these larger architectures to ensure optimal user experience. Following your suggestion, we conducted additional effectiveness experiments on models with varying parameter scales (ranging from 2B to 38B), maintaining the experiment settings described in Section 5.3. We report the CGSR results below (noting that Rank metrics are excluded as they depend solely on the retriever).
> | Models | Qwen3-VL-2B-Instruct | InternVL3-2B | Qwen3-VL-32B-Instruct | InternVL3.5-38B |
> | :--- | :--- | :--- | :--- | :--- |
> | AQUA_acronym CGSR | 92.18% | 70.53% | 99.83% | 90.71% |
> | AQUA_spatial CGSR | 91.67% | 63.76% | 98.91% | 84.33%|
>
> The experimental results demonstrate the robustness of our method across different model sizes. Specifically:
> - **Low-resource robustness:** Even the smallest model, InternVL3-2B, achieves a CGSR exceeding 60%, which provides a sufficient margin for effective watermark detection. Notably, Qwen3-VL-2B-Instruct outperforms the 7B models previously reported in the paper, despite its significantly smaller parameter size.
> - **Scalability:** As anticipated, models in the 30B parameter range consistently outperform their 7B counterparts.
> These findings confirm that AQUA is not constrained by model size and generalizes effectively across different parameter scales.
>
> > **Q2:** Does not assess how watermark detection performance degrades over time with retriever/generator updates, fine-tuning, or dataset drift.
>
> Our method (AQUA) fundamentally relies on the retriever to perform image-text embedding similarity search, and the generator to comprehend visual content and conduct reasoning following user instructions. These core capabilities are intrinsic to the pre-trained models and are inherently robust; they do not degrade easily, even under fine-tuning or when adapting to different scenarios. To empirically validate this robustness, we conducted the following experiments:
> 1. **Generalization across Retrievers:** We replaced the retriever with SigLIP (siglip-so400m-patch14-384) and evaluated it on the MMQA dataset using the same experiment settings as Table 1 in Section 5.3 Effectiveness of AQUA. As shown below, the results are highly consistent with the original CLIP-ViT model(clip-vit-large-patch14-336), demonstrating that our method generalizes well across different retriever models.
>
>     | Metric | AQUA_acronym | AQUA_spatial |
>     | --- | --- | --- |
>     | Rank | 1.106 | 1.386 |
>
> 2. **Robustness under Retriever Fine-tuning:** To simulate a realistic adaptation scenario, we fine-tuned the clip-vit-large-patch14-336 model using 1,000 image-question pairs (rather than image-caption pairs) from the MMQA dataset for 1 epoch (learning rate: 1e-6). The results indicate that the ranking metrics remain stable after fine-tuning.
>
>     | Metric | AQUA_acronym | AQUA_spatial |
>     | --- | --- | --- |
>     | Rank | 1.038 | 1.376 |
>
> 3. **Robustness against Fine-tuning:** We further evaluated the robustness of our method when the generator undergoes fine-tuning. Specifically, we fine-tuned the Qwen2.5-VL-7B-Instruct model using LoRA on a diverse mixed dataset composed of COCO (image captioning, 500 samples), and Alpaca-GPT4 (general instruction following, 300 samples).
>
>     We configured LoRA to target all linear layers with a learning rate of 3e-5 to ensure deep parameter updates. As shown in the table, the CGSR metrics exhibited only a marginal decline, confirming our method's robustness despite the significant shift in the model's distribution.
>
>     | Metric | CGSR (%) |
>     | :--- | :--- |
>     | AQUA_acronym | 97.92% |
>     | AQUA_spatial | 96.09% |
>
> 4. **Robustness with Large-Scale Dataset Drifts:** Finally, we tested the robustness of our approach against varying sizes and contents of the retrieval corpus. Based on the original MMQA dataset(~58k images), we progressively added 10k to 50k images from the WebQA dataset as distractors. We used clip-vit-large-patch14-336 as the retriever and Qwen2.5-VL-7B-Instruct as the generator. The experiment results demonstrate that even with a significant increase in dataset size (nearly doubled), the performance remains stable, highlighting the scalability of our method.
>
>     | Added Images | +10k | +20k | +30k | +40k | +50k |
>     | :--- | :--- | :--- | :--- | :--- | :--- |
>     | AQUA_acronym Rank | 1.026 | 1.028 | 1.031 | 1.029 | 1.042 |
>     | AQUA_spatial Rank | 1.351 | 1.526 | 1.424 | 1.496 | 1.696 |
>     | AQUA_acronym CGSR| 98.77%| 98.39% | 96.10% | 97.21% | 96.76% |
>     | AQUA_spatial CGSR| 96.19% | 93.81%| 92.56% | 95.77% | 91.85%|

---

> > ### Author Response · Authors · 2025-11-23
> >
> > > **Q3:**  While mentioning a reference distribution for practical verification, it provides only a single example without guiding how to adapt it to diverse dataset characteristics or RAG system configurations.
> >
> > Establishing a reference distribution is a standard practice in watermarking research, as demonstrated in prior works [1], [2], and [3]. Since verification distributions vary across different datasets and watermark patterns, ensuring rigorous verification requires data providers to derive distributions specific to their datasets. The reference distribution presented in our original manuscript is specific to the MMQA dataset.
> > To facilitate application to other datasets, data providers can generate their own reference distributions using the following procedure:
> > 1. **Select a RAG System:** Employ an arbitrary RAG system (either open-source or closed-source).
> > 2. **Clean Distribution:** Use the non-watermarked dataset as the knowledge base. Following the watermark verification protocols in Sections 4.1 and 4.2, perform probe queries to obtain the VSR metrics, calculating the mean and variance.
> > 3. **Watermarked Distribution:** Replace the knowledge base with the watermarked dataset and repeat the process to obtain the second distribution.
> >
> > To demonstrate the robustness of our method across different settings, we have conducted additional experiments using the MMQA and WebQA datasets and alternative RAG system configurations. As shown in the table below, regardless of the specific RAG system or dataset employed, there remains a significant divergence between the clean (non-watermarked) and watermarked distributions. This distinct separation ensures reliable verification across diverse environments.
> > | RAG and Watermark Setting | Dataset| $μ_{clean}$ | $σ_{clean}^2$ | $μ_{wm}$ | $σ_{wm}^2$ |
> > | --- | --- | --- | --- | --- | --- |
> > | LLaVA-NeXT_acronym | MMQA | 0.01 | 0.02 | 0.60 | 0.20 |
> > | LLaVA-NeXT_spatial | MMQA  | 0.20 | 0.20 | 0.55 | 0.25 |
> > | InternVL3_acronym | MMQA  | 0.01 | 0.01 | 0.86 | 0.13 |
> > | InternVL3_spatial | MMQA  | 0.12 | 0.18 | 0.75 | 0.21 |
> > | Qwen2.5-VL-Instruct_acronym | MMQA  | 0.00 | 0.01 | 0.98 | 0.02 |
> > | Qwen2.5-VL-Instruct_spatial | MMQA  | 0.07 | 0.13 | 0.97 | 0.09 |
> > | LLaVA-NeXT_acronym | WebQA | 0.01 | 0.02 | 0.77 | 0.03 |
> > | LLaVA-NeXT_spatial | WebQA | 0.04 | 0.11 | 0.86 | 0.15 |
> > | InternVL3_acronym | WebQA | 0.01 | 0.01 | 0.79 | 0.19 |
> > | InternVL3_spatial | WebQA | 0.12 | 0.18 | 0.76 | 0.23 |
> > | Qwen2.5-VL-Instruct_acronym | WebQA| 0.00 | 0.01 | 0.96 | 0.05 |
> > | Qwen2.5-VL-Instruct_spatial | WebQA| 0.07 | 0.13 | 0.89 | 0.10 |
> >
> > It is important to note that the reference distribution parameters presented in the original manuscript were established as conservative estimates, with intentionally broadened variances to accommodate potential system fluctuations in real-world deployments. In contrast, the table above presents precise empirical statistics derived directly from our additional experiments. As evident from the data, the actual separation between the clean and watermarked distributions is consistently distinct across all tested models and datasets, confirming the robustness of our verification mechanism.
> >
> > [1] WARD: Provable RAG Dataset Inference via LLM Watermarks
> >
> > [2] DMI-RAG: Dataset Protection via Watermarked Canaries in Retrieval-Augmented LLMs
> >
> > [3] Towards Copyright Protection for Knowledge Bases of Retrieval-augmented Language Models via Reasoning
> >
> > We sincerely appreciate your thorough review and for highlighting these important points. Your valuable feedback will certainly help us enhance our methodology and improve the clarity of our presentation. Please don't hesitate to reach out if you have any further questions or would like to discuss this further.

---

> ### Author Response · Authors · 2025-11-27
> **Inquiry regarding remaining concerns**
>
> Dear Reviewer k6AT,
>
> I hope this message finds you well.
>
> As the discussion period is nearing its end with **less than one week remaining**, I wanted to ensure that we have addressed all your concerns satisfactorily. If there are any additional points or feedback you would like us to consider, please let us know.
>
> If you feel that our responses and revisions have effectively resolved your concerns, we would be grateful if you could take these improvements into account for your final assessment.
>
> Thank you again for your time and effort in reviewing our paper.
>
> Best regards,
>
> The Authors

---

### Author Response · Authors · 2025-11-30
**Summary of Discussion Improvements and Methodological Strengths**

Dear Area Chair and Reviewers,

In line with the expectations of the ICLR organizing committee, we have included this section to summarize our replies to the reviewers’ comments. We believe that the updated manuscript, together with the clarifications provided during the discussion phase, has addressed all of the concerns raised and now adequately satisfies the reviewers’ requirements.

We have offered point-by-point responses to each reviewer in their respective sections. Below, we summarize how our clarifications and additional experiments during the discussion phase have addressed these concerns and further substantiated the **methodological strengths** of AQUA:

1. **Generalizability across Models and Settings:** We verified that AQUA works effectively on both lightweight (2B) and large-scale (30B+) VLMs, ensuring consistent performance regardless of model size (Response to k6AT Weakness 1). To support its use in different RAG systems, we added a clear explanation of how the reference distribution is collected and applied (Response to k6AT Weakness 3). We also clarified that AQUA_acronym and AQUA_spatial handle different scenarios, meaning they can be deployed together to cover more cases (Response to vGro Question 1).

2. **Robustness to System Updates and Adaptive Attacks:** We showed that AQUA remains effective throughout the RAG lifecycle, even after fine-tuning the retriever/generator or adding new data to the knowledge base (Response to k6AT Weakness 2). Our experiments further confirm that current SOTA models readily accept our watermark images, and we found that trying to train the model to ignore these watermarks will certainly damage its general visual reasoning capability (Response to DCH9 Weakness 2, Question 2; vGro Weakness 2).

3. **Practical Reliability and Rigorous Design:** We clarified that AQUA inherently handles potential watermark collisions through statistical verification, and we added a discussion on adopting a centralized management approach in the revised paper to rule out conflicts entirely (Response to DCH9 Weakness 1, Question 1). We provided a detailed breakdown of the pipeline used to ensure the semantic uniqueness of our triggers. Furthermore, we clarified how to interpret the $Rank$ metric in Table 2 (Response to DCH9 Weakness 3, Question 3; vGro Question 4). Finally, we articulated AQUA’s distinct advantages over traditional watermarking and detailed how the method can be effectively extended to handle copyright attribution (Response to vGro Question 2, 3), while also justifying our choice of baselines (Response to vGro Weakness 3).


We sincerely thank all reviewers for their valuable feedback. We are pleased our contributions were well received and appreciate recognition of our work’s strengths:

1. **Novel Problem Formulation:** Reviewers recognized our work as the first to formally address image copyright in Multimodal RAG. They specifically appreciated the identification of "indirect watermark propagation" as a core challenge and a key contribution to image data governance. [k6AT, DCH9, vGro]

2. **Creative and Practical Design:** Our semantic-based approach (utilizing acronyms and spatial relationships) is considered a creative solution that works across modalities. Reviewers noted its adaptability to both black-box and white-box scenarios, establishing a strong baseline for real-world RaaS applications. [k6AT, DCH9, vGro]

3. **Robustness and Effectiveness:** The evaluation is acknowledged as thorough, covering multiple datasets and models. Reviewers highlighted AQUA’s robustness against various image transformations (e.g., rotation, compression) and attacks, while remaining imperceptible to users. [k6AT, vGro]

4. **Clarity of Presentation:** The paper is praised for being well-structured and easy to follow. Reviewers pointed out that the visualizations (e.g., Figures 1-3) and precise definitions significantly aided the understanding of the methodology. [DCH9, vGro]

We sincerely hope that our revisions and responses have adequately addressed the concerns raised. We are deeply grateful for the constructive feedback, which has been instrumental in improving our manuscript. We have included this summary to provide a clear overview for the Area Chair’s convenience during the final assessment.

Sincerely,

The Authors

---

### Meta-Review · Area_Chair_dnaB · 2025-12-28

**Summary:**

This paper proposes AQUA, a framework for copyright protection in multimodal Retrieval-Augmented Generation (RAG) models. The main idea is embedding semantic signals into images through two complementary watermarking techniques: AQUAacronym (embedding uncommon acronyms and their full names) and AQUAspatial (using spatial relationships in the image).  Experiments show that AQUA is robust, stealthy, and effective in tracing copyright.

Strength of the paper:
1. The watermarking techniques, particularly AQUAacronym and AQUAspatial, are shown to be robust against various image transformations and attacks.
2. The experiments show that the AQUA method is effective.

Weakness of the paper:
1. the paper lacks generalization of the method on larger models. (authors provided additional results on both lightweight and large models in response).
2. the paper does not discuss performance degrades over time with retriever/generator updates, fine-tuning, or dataset drift. ( the authors addressed these with additional experimental results on retriever update, fine-tuning, dataset shift).
3. risk of collision of watermarks. (the author addressed this with additional explanation).

Reviewer vGro's review is very shot and not concrete though the authors still provided reasonable response. Reviewer vGro raised the question about extension to attribution problem in "Watermark-based Attribution of AI-Generated Content", but the question is irrelevant to the problem studied in this paper and is therefore disregarded in the assessment.
However, the author should not modify the original template provided by the conference (especially they should not reduce the space between paragraphs. please do not do this next time).

Overall, the paper proposed an effective method for watermarking VLLM RAG models. The authors addressed concerns well.

**Reviewer Concerns:**

All major concerns are addressed.

**Reviewer Scores:**

Yes. Both reviewers giving lower scores would likely to raise the scores based on the author responses.

---

### Decision · Program_Chairs · 2026-01-26

Accept (Poster)